# Axial localization and tracking of self-interference nanoparticles by lateral point spread functions

Yongtao Liu [1,11], Zhiguang Zhou [1,11], Fan Wang [1,2,11✉], Günter Kewes[3], Shihui Wen [1], Sven Burger [4,5], Majid Ebrahimi Wakiani[1,6], Peng Xi[1,7,8], Jiong Yang[1,9], Xusan Yang [7,10], Oliver Benson[3✉] & Dayong Jin [1,8✉]

Sub-diffraction limited localization of fluorescent emitters is a key goal of microscopy imaging. Here, we report that single upconversion nanoparticles, containing multiple emission centres with random orientations, can generate a series of unique, bright and position-sensitive patterns in the spatial domain when placed on top of a mirror. Supported by our numerical simulation, we attribute this effect to the sum of each single emitter's interference with its own mirror image. As a result, this configuration generates a series of sophisticated far-field point spread functions (PSFs), e.g. in Gaussian, doughnut and archery target shapes, strongly dependent on the phase difference between the emitter and its image. In this way, the axial locations of nanoparticles are transferred into far-field patterns. We demonstrate a real-time distance sensing technology with a localization accuracy of 2.8 nm, according to the atomic force microscope (AFM) characterization values, smaller than 1/350 of the excitation wavelength.

[1] Institute for Biomedical Materials and Devices (IBMD), Faculty of Science, University of Technology Sydney, Sydney, NSW 2007, Australia. [2] School of Electrical and Data Engineering, Faculty of Engineering and Information Technology, University of Technology Sydney, Ultimo, NSW 2007, Australia. [3] AG Nanooptik, Institut für Physik & IRIS Adlershof, Humboldt Universität zu Berlin, Newtonstraße 15, 12489 Berlin, Germany. [4] JCMwave GmbH, Bolivarallee 22, 14050 Berlin, Germany. [5] Zuse Institute Berlin, Takustraße 7, 14195 Berlin, Germany. [6] School of Biomedical Engineering, Faculty of Science, University of Technology, Sydney, NSW 2007, Australia. [7] Department of Biomedical Engineering, College of Engineering, Peking University, Beijing 100871, China. [8] UTS-SUStech Joint Research Centre for Biomedical Materials & Devices, Department of Biomedical Engineering, Southern University of Science and Technology, Shenzhen, Guangdong 518055, PR China. [9] School of Chemical Engineering, University of New South Wales (UNSW), Sydney Campus, Sydney, NSW 2052, Australia. [10] School of Applied and Engineering Physics, Cornell University, Ithaca, NY 14853, USA. [11] These authors contributed equally: Yongtao Liu, Zhiguang Zhou, Fan Wang. ✉email: fan.wang@uts.edu.au; oliver.benson@physik.hu-berlin.de; dayong.jin@uts.edu.au

Super-resolution localization of fluorescent emitters, particularly along the axial dimension, poses a key challenge in optical microscopy. Approaches based on excitation confinement are relatively straightforward. Total internal reflection fluorescence microscopy confines the excitation within a thin region above the substrate surface to achieve an axial resolution of 100–200 nm[1]. Single plane illumination microscopy uses the narrower lateral section of the excitation beam to achieve optical sectioning perpendicular to the axial direction, providing an axial resolution of 450–750 nm[2]. Axial interference is an efficient way to confine the excitation region and improve the axial resolution. I[5]M microscopy uses two opposing objectives with two incoherent illumination beams to interfere each other, together with image interference, so as to achieve 100 nm axial resolution[3]. Similarly, 4Pi microscopy uses the optical interference of two excitation focal spots to achieve an axial resolution of 75 nm[4]. Placing a mirror after the focal point will generate an isotropic excitation spot, due to the interference between the excitation and its reflection[5], which confines the confocal PSF into a 110-nm spot[6]. Similar to the strategy by confining the excitation region, the confinement of emission regions also enhances the axial resolution. Supercritical angle fluorescence microscopy detects the emission above the critical angle of a medium interface, which creates an emission collection region of 100–200 nm above the interface. By applying an axial hollow depletion beam to deactivate the fluorophores near the focal point, stimulated emission depletion microscopy can achieve 150 nm axial resolution[7].

Approaches by encoding the axial position information into the time domain are indirect ways, but often with higher axial localization accuracy. For example, in the time domain, metal- or graphene-induced energy transfer (MIET or gMIET) approaches rely on the lifetime measurement that is steeply dependent on distance, when fluorescent molecules are nearby a metal[8] or graphene surface[9]. These methods provide an axial resolution from a few nanometers to the ångström level. However, the achievable localization accuracy is limited by the range of energy transfer interaction (L), following the relationship $L/\sqrt{N}$, where N is the budget of photons from a single molecule. For example, for MIET using gold or silver films, L ranges from ~150 to 200 nm, so that the localization accuracy for single molecules is around 5–6 nm at a photon budget of 1000 photons; for gMIET using a graphene substrate, L is reduced to ~20 nm, which explains why sub-nanometer localization accuracy can be achieved. These energy transfer-based technologies require the measuring sample closing to the substrate surface (e.g., <200 nm).

Approaches by encoding axial position information in the spectral domain, through emission's self-interference (SELFI) effect, can provide a long axial sensing range. As spontaneous emission has a low degree of coherence, external references cannot be used, and therefore SELFI is the only way to manage the interference for the incoherent fluorescence. Spectral SELFI microscopy uses both the reflecting mirrors and large spacers (>10λ) to decode the vertical position of a fluorescent layer with high accuracy of 0.2 nm, achieved by measuring the spectrum fringes[10,11]. Mock et al. locate a gold nanoparticle above a metallic substrate to selectively enable the surface plasmon polariton and the localized surface plasmon. This, in turn, provides a distance-dependent spectrum profile, resulting in an axial resolution <1 nm[12]. While this method is restricted to metallic nanoparticle, substrate and the total internal reflectance illumination, its sensing range is limited to smaller than 100 nm. Transferring the axial information into the spectrum or time domain typically requires sophisticated time-resolved or spectra-analysis systems, which often limits the sensing speed for the 2D lifetime or spectrum mappings.

The axial position information can also be encoded into the optical phase variation. Shtengel et al. used a multi-path emission interferometric photoactivated localization microscopy to retrieve the position information from the phase information of the emitting photons, and achieved a sub-20 nm axial resolution[13]. Jouchet et al. developed a modulated localization (ModLoc) strategy using the amplitude-modulated LIDAR approach, and achieved a 7.5 nm axial resolution[14]. ModLoc is based on a lock-in four-imaging-channel system with interference excitation patterns to retrieve the phase information of the emitter. These methods require both the complicated interferometric arrangement and the phase retrieving systems.

Approaches based on emission PSF engineering can detect the axial position for a larger range. The PSF can be modulated by defocusing[15], optical astigmatism[16], and spatial light phase mask[17,18]. The localization can be achieved by measuring the PSF profile change. The recently reported approach, SELFI microscopy generates fluorescence SELFI within each detected PSF, so that in a single camera frame, parallel analysis of the quantitative fluorescence intensity and wavefront curvature can lead to both lateral and axial localization of single fluorescent molecules[19] and video-rate tracking of quantum dots[20]. As these methods are too sensitive to the focus position, the focus drifting will affect the accuracy. Plasmonic coupling-induced polarization selection can also modify the PSF, however, the working range is limited (e.g., <40 nm)[12].

Here, we present a concept for a single nanoparticle to achieve SELFI. We demonstrate how to transfer the axial position of a nanoparticle into the far-field lateral PSF, which enables axial distance sensing from 2D imaging to substantially simplify the sensing process. When placing the nanoparticle on a mirror, each of the emitted field components interferes with its reflected image and creates a characteristic far-field pattern that varies strongly with the phase difference between the emitter and its image. SELFI arises from superposing radiating dipoles with their own emission reflected from the mirror. To the best of our knowledge, the spatial distribution of the spontaneous emission's SELFI from multiple emitters at the nanoscale has not been reported. Such a kind of SELFI leads to a fast, high-resolution, and anti-drift sensing method to accurately resolve the position of a single nanoparticle along the axial axis, and most importantly it is suitable for conventional widefield fluorescence microscopy setups without system modification requirement.

## Results

**Multi-emitters in lanthanide-doped upconversion nanoparticles (UCNPs).** To verify our approach, we choose lanthanide-doped UCNPs, co-doped with a network of multiple sensitizer and emitter ions[21]. In the typical process of photon upconversion, sensitizer ions (e.g., $Yb^{3+}$) absorb the lower level energy excitation photons at near-infrared and transfer the sensitizer energy to their nearby emitter ions (e.g., $Tm^{3+}$). It is the emitter ion with metastable intermediate states that can nonlinearly upconvert the energy onto the higher levels and eventually emit the anti-Stokes luminescence. In such a nanoscale system with a network of at least thousands of sensitizers and hundreds of emitters, the orientation of emitting dipoles are independent of the excitation field. A single UCNP can be highly doped with more than $10^4$ ions, with each ion's 4f atomic orbital being shielded[22], to generate high brightness[23–25], non-blinking and photo-bleaching resistant emissions. Their exceptional brightness and non-linear optical properties make UCNPs suitable as probes for single-molecule imaging[26–28] and super-resolution imaging[29,30]. UCNPs can be a perfect model system to investigate multi-emitters' SELFI, as they contain a dense cluster

of isolated dipole emitters within a diffraction-limited spot. According to Borel's law of large numbers, the emitters are randomly oriented to provide emissions from all three orthogonal dipole orientation components with statistically equal amplitude. Moreover, lanthanide ions have distinct narrow emission bands in their spectrum even at ambient conditions, which is favorable for optical SELFI that requires the caption of narrowband emission (see Supplementary Information Note 1).

**Mirror induced emission self-interference effect.** As shown in Fig. 1a, placing single UCNPs on a mirror substrate with a layer of SiO₂ provides a defined phase difference between the direct emission and the emission reflected by the mirror. Though the electric field distribution radiated from a dipole near a substrate is well developed[31], its far-field PSF is still challenged to be expressed as a mathematic formula. In order to understand the distance-dependent PSF under SELFI, we conducted a numerical simulation by finite element analysis (JCMwave, details in Supplementary Information Note 3). We calculated the far-field image pattern for an emission wavelength of 455 nm. Each UCNP contains many emitters that can be represented as emitting dipoles with random oscillation orientations. Statistically, the oscillation components of dipoles along the $x$-, $y$-, and $z$-axis should be equally weighted. Hence, in the simulation, we first use three emitting dipoles as sources with oscillation orientation at $x$-, $y$- and $z$-axis ($x$-dipole, $y$-dipole, and $z$-dipole) and incoherently add their emission. The 3D SELFI pattern of each dipole in the far-field is shown in Fig. 1a(i–iii). Two-dimensional cuts ($x$-$y$ plane) at $z = 0$ for dipoles oriented along the $x$-, $y$-, and $z$-axes are similar to the traditional PSF for dipoles without the SELFI between their emission and the reflected emission[32]. However, in striking contrast to the traditional emission without SELFI, both $x$-dipole and $y$-dipole feature have a local intensity minimum along the $z$-axis, as shown at the $z$-$x$ plane in Fig. 1a(i) and Fig. 1a (iii), respectively. The intensity minimum results from a destructive SELFI for each dipole with a spacing of 154.9 nm to the mirror surface. The $z$-dipole does not have a minimum along the $z$-axis, as its emission does not propagate along the $z$-axis. The total emission field from a nanoparticle is now calculated as the incoherent sum (with equal weights) of the SELFI field of $x$-, $y$-, and $z$-dipoles, as all spontaneous emission processes are expected to be uncorrelated. Figure 1b reveals a hollow ellipsoid three-dimensional shape, where its dark centre is due to destructive interference.

The simulated PSF also indicates a doughnut PSF at $x$-$y$ plane, which matches with the experimental pattern (Fig. 1c) that is measured by widefield fluorescence image of UCNPs on mirror substrate (spacing is 154.9 nm). As the confocal microscopy confines the excitation PSF and the detection PSF through a pinhole, rather than the widefield emitting PSF, here we employ the simple widefield microscope (Supplementary Fig. 4) with the 980 nm excitation to image the spatial distribution of the SELFI induced PSF. As expected, by omitting SELFI the PSF of UCNPs on the SiO₂ substrate alone shows a standard Gaussian distribution (Fig. 1d). We further confirm the SELFI effect by measuring the UCNPs' fluorescence spectrum. Figure 1e shows the emission spectrum from a UCNP in the interferometric configuration. Compared to the emission spectrum from a UCNP on SiO₂ substrate alone (Fig. 1f), there is a dramatic modification, in particular a decrease in 455 nm and an increase in the near-infrared band (700–800 nm). The spectral features thus confirm the occurrence of pronounced interference effects. The lifetime measurement results (Supplementary Fig. 7) confirm that the plasmonic quenching is negligible when the working range is larger than 75 nm. The numerical simulation of energy transfer

(see Supplementary Information Note 4) indicates a small emission enhancement due to the plasmonic coupling, which has been considered during the simulation process. Notably, this characteristic feature produced by the SELFI effect is robust as it also appears at other wavelengths (see Supplementary Figs. 12 and 13).

**Phase-dependent self-interference.** Quantitative modeling of the observed self-interferent emission PSF is a multi-parameter problem. The accuracy will depend on the parameters, such as the wavelength, the emitter-to-mirror distance (e.g., modified Purcell factor and angular emission), the immersion media, the size of the nanoparticle (spatial coherence), the detected spectrum (temporal coherence), the quantum yield of the emitters, the excitation condition (e.g., beam profile and intensity), the optics setup (e.g., the numerical aperture and system aberration) and the entire photon upconversion process. Here we employ a simple model, three co-localized orthogonal dipoles radiating in front of a mirror with a single wavelength of emission, to capture the relevant physics, as this model can semi-quantitatively reproduce our experimental observations. We take this to illustrate the general applicability of the SELFI concept. For the in-depth discussions of the aforementioned effects, see Supplementary Information Note 4.

We now provide an analytic approach to give an idea of the behavior of the PSF patterns along the $z$-axis. The emission from a nanoparticle propagates to the mirror, is reflected and returns to the nanoparticle to interfere with the unreflected emission. The propagation distance induces a phase change ($\phi_Z$). The field reflection of the emission at the interface between the beam propagating medium (SiO₂) and the silver (Ag) surface induces an external phase shift $\phi_{refl}$, with an expression of $\phi_{refl} = \pi - \tan^{-1}(\mathrm{imag}(\widetilde{r})/\mathrm{real}(\widetilde{r}))$; $\widetilde{r} = (\widetilde{n} - \widetilde{n}_{ag})/(\widetilde{n} + \widetilde{n}_{ag})$ for a normal incident angle, according to the Fresnel law. Here $\widetilde{n}$ and $\widetilde{n}_{ag}$ are the complex refractive index of the propagating medium and the silver mirror, respectively. For the nanoscale emitter, the Gouy phase $\phi_{Gouy}$ has to be taken into account. The phase difference induced by the curvature of the emission's wavefront is negligible here, due to the interference happens around the axial axis. Hence the total phase difference can be written as:

$$\Delta\phi(\Delta z) = \phi_z(\Delta z) + \phi_{\mathrm{refl}} + \phi_{\mathrm{Gouy}}(\Delta z) \quad (1)$$

where $\phi_z = 4\Delta a\pi n/\lambda$, $\lambda$ is the emission wavelength, $\Delta z$ is the distance between the UCNP and the mirror surface. The Gouy phase shift $\phi_{Gouy}$ is given by[33]:

$$\phi_{\mathrm{Gouy}}(z) = \tan^{-1}\frac{2\Delta z}{z_R} \quad (2)$$

where $z_R$ is the Rayleigh range of the emission beam. Notably, the Gouy phase shift plays an important role here, for instance, a 154.9 nm distance will induce a phase shift of 0.23$\pi$, which modifies the interference type. According to Eq. (1), complete destructive interference and constructive interference occur when $\Delta\phi = (j + 1)\pi$ and $\Delta\phi = j\pi$ respectively, $j$ is an integer. These considerations match very well with the experimental results. For instance, when the spacing is around 71.6 and 154.9 nm, the total phase difference is 1.9$\pi$ and 3.1$\pi$, resulting in constructive interference (Fig. 2a$_i$, g$_i$) and destructive interference (Fig. 2c$_i$, i$_i$), respectively. Note that the mirror will enhance the emission intensity by reflecting the otherwise undetected emission in a forward direction. However, the enhancement depends on the spacing distance, as the destructive interference may redirect more emissions beyond the maximum collecting angle given by the NA of the optical system

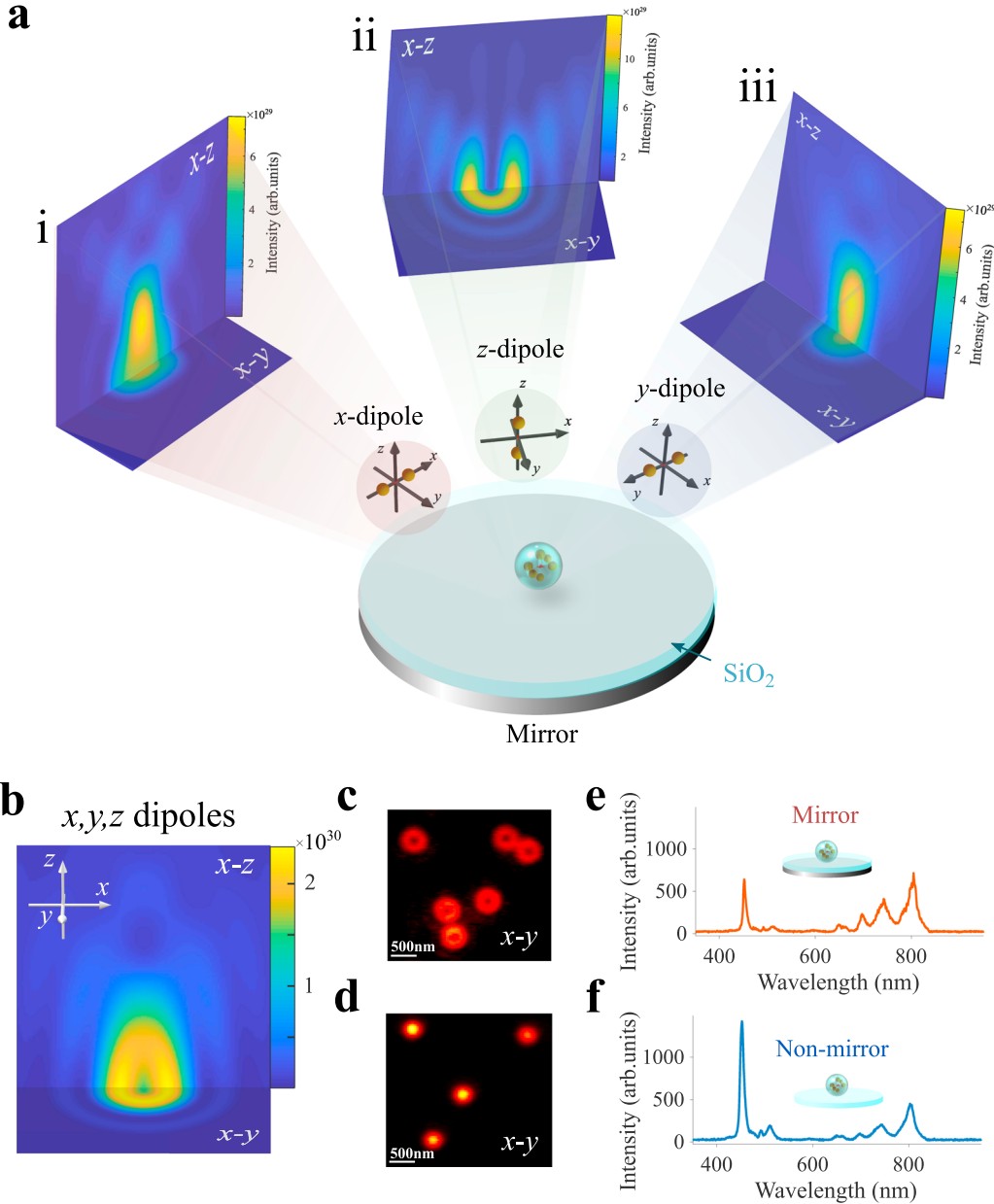

**Fig. 1 Spatial self-interference of UCNPs on a mirror substrate. a** A single UCNP is placed on a mirror substrate with a silica layer as the spacer. Calculations of self-interference of dipole emitters with different orientations on a mirror substrate were performed with a finite element Maxwell's equation solver (JCMwave) for a $SiO_2$ spacer layer with a spacing of 154.9 nm. (i), (ii), and (iii) show different cuts through the far-field PSFs of self-interference patterns of oriented dipoles oscillating along the $x$-, $y$-, and $z$-axis, respectively. The horizontal two-dimensional cuts are for $x$-$y$ planes with $z = 0$ (dipoles' centre). The vertical cuts are for $x$-$z$ planes with $y = 0$ (dipoles' centre). The emission wavelength in simulation is 455 nm. **b** The total emission field from the self-interference field of $x$-, $y$-, and $z$-dipoles. In the simulation for **a** and **b**, the origin ($x = 0$, $y = 0$, $z = 0$) of the coordinate system is in the centre of the UCNP, while the Ag mirror is located 154.9 nm below the origin. Experimentally, the weights of the self-interference field of $x$-, $y$-, and $z$-dipoles will be modified by the effective numerical aperture of the imaging collecting system, which has been considered in the simulation. **c** Widefield fluorescence image of UCNPs in the configuration with a UCNP-to-mirror distance (spacing) of 154.9 nm. **d** Widefield fluorescence image of UCNPs on a cover glass surface. The widefield fluorescence image of UCNPs is filtered through a blue color filter (475 ± 12.5 nm). A defocused 980 nm laser creates the widefield excitation. The scale bar in **c** and **d** is 500 nm. The measured emission spectrum from a single UCNP on **e** a mirror substrate with a spacing of 154.9 nm and **f** a cover glass surface. The UCNPs used in the experiment are β-NaYF₄:20%Yb³⁺,8%Tm³⁺, with a diameter of 33 nm. The size of UCNPs is characterized by transmission electron microscopy, as shown in Supplementary Fig. 2. (Further details on the synthesis and characterization of UCNPs as well as fabrication of the mirror substrate are provided in Supplementary Information Note 2). The spectra are measured by a single-particle characterization system described in detail in ref. [44].

(see Supplementary Fig. 14 for the details of the emission intensity variations).

Figure 2a$_i$–f$_i$, g$_i$–l$_i$ shows the numerical simulation of PSF at the $x$-$y$ plane ($z = 0$) and the $y$-$z$ plane ($x = 0$), respectively. The nanoparticles locate at the measuring focal plane where $z = 0$. As the distance increases, the PSF changes periodically from constructive to destructive interference. Figure 2a$_{ii}$–f$_{ii}$, g$_{ii}$–l$_{ii}$ shows the experimental results of measured PSFs. These experimental results qualitatively match the numerical simulation, further confirming our interpretation of nanoscale SELFI.

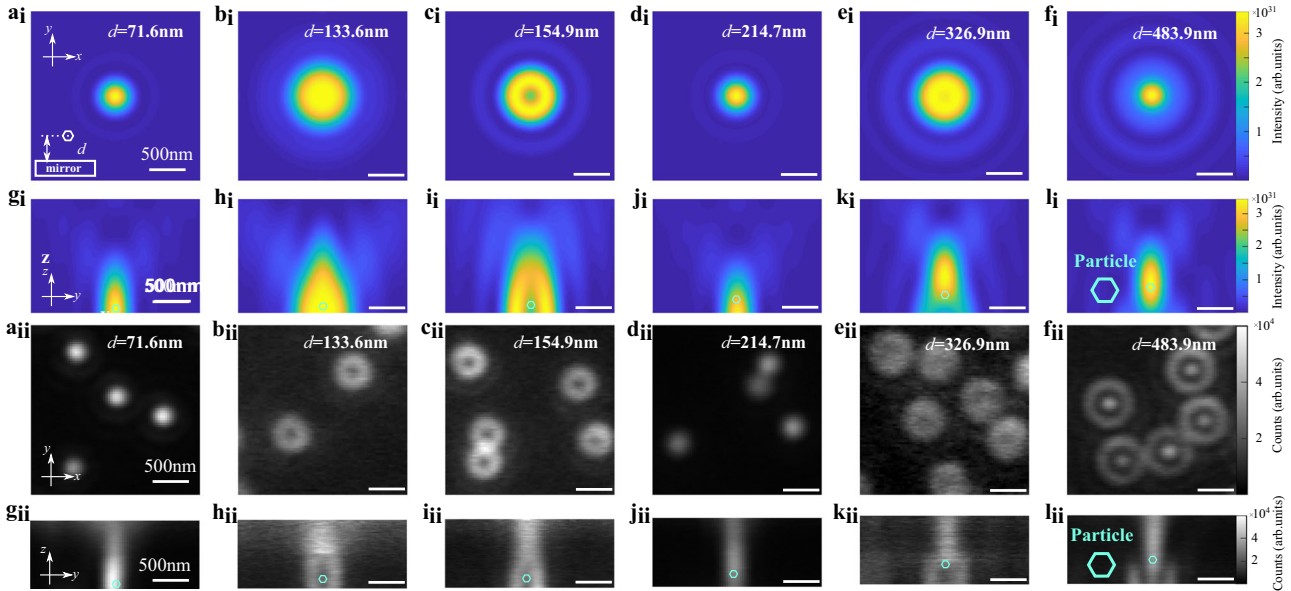

**Fig. 2 Comparison of simulated and experimental distance-dependent far-field PSFs of UCNPs' emission self-interference.** The UCNP-to-mirror distance $d$ (see inset in $a_i$) increases from the left to the right column as indicated from 71.6 to 483.9 nm. The simulated PSFs of the self-interference for a particle on a mirror substrate with spacing distances of 71.6, 140, 154.9, 214.7, 326.9, and 483.9 nm, at the $x$-$y$ plane ($a_i$–$f_i$) and $y$-$z$ plane ($g_i$–$l_i$). The experimentally measured PSFs of the upconversion emission self-interference for UCNPs on a mirror substrate with the distance of 71.6, 133.6, 154.9, 214.7, 326.9, and 483.9 nm away from the mirror surface, at the $x$-$y$ plane ($a_{ii}$–$f_{ii}$) and $y$-$z$ plane ($g_{ii}$–$l_{ii}$). $a_{ii}$–$f_{ii}$ Recorded by imaging the UCNPs at the $x$-$y$ plane. $g_{ii}$–$l_{ii}$ Recorded by stacking defocused $x$-$y$ plane images. The radius of the UCNPs was 16.5 nm, and the emission wavelength used for detection is at 455 nm. A defocused 980 nm laser creates the widefield excitation. More experimental and simulation results for 455 and 800 nm emission wavelengths are shown in Supplementary Figs. 11 and 12, respectively. The scale bar is 500 nm. The imaging focal plane in $a_i$–$f_i$ and $a_{ii}$–$f_{ii}$ is at $z = 0$, where the UCNP is located. Experimentally the $z = 0$ plane is determined by moving the focus above the mirror surface with spacing distance. The $z$ position of the mirror surface is located by the reflection of a Gaussian beam. The tiny cyan hexagon indicates the $z$-$y$ plane position of the UCNPs, where $z = 0$ and $y = 0$. The experiment results were captured by an EMCCD with an exposure time of 20 ms. The experimental PSFs at the $z$-$y$ plane is the cross-section images of a 3D volume image generated by stacking 160 defocused images (20 nm for step size) at the $x$-$y$ plane. The defocusing is well-defined and achieved by the objective's piezoelectric positioner. The surrounding medium of UCNPs is the immersion oil with a refractive index of 1.5. All scale bars in this figure are 500 nm.

The experimental result shows a longer depth of field at the $z$-axis distribution of the intensity, which is due to the non-parallel excitation of the widefield beam. The slight discrepancy of 2D PSF shapes between the experimental and the simulated results are caused by the system aberration. Under constructive interference condition, the SELFI for dipoles with $x$- and $y$-orientation generates a bright spot at the middle position (Fig. $2a_{ii}$, $g_{ii}$), while with destructive interference, they generate a dark spot (Fig. $2c_{ii}$, $i_{ii}$) in both 2D and 3D PSF. Notably, Figure $2e_{ii}$, $f_{ii}$ resembles defocused emission patterns. However, the defocusing of far-field emission pattern from a point source cannot generate the unique semi-ellipsoid dark area in the 3D PSF[34,35] (Fig. $2h_{ii}$).

Not only is the narrow emission bandwidth crucial for nanoscale SELFI, the small size of UCNPs with high-density emitters is also mandatory, as larger structures would cause a blurring of the interference patterns. The average phase difference of light emitted from two emitters in a UCNP with a radius of 16.5 nm is only $\phi_{blur} = 2\pi \frac{R}{\lambda} n \sim \pi/10$ ($\lambda = 455$ nm, $R$ is the average distance[36]), which does not prevent the observation of SELFI. Besides, the larger size of particles will result in a wider distribution of emitters along the $z$-axis, and thereby decreasing the gradient of the calibration curve (see Supplementary Information Note 4(3)).

mirror surface. We use four parameters to distinguish different PSFs: (1) the outside full width at half maximum (Fig. 3g); (2) the full width at half maximum of the inner dip (Fig. 3h); (3) the area of the averaged cross-section (Area, Fig. 3i); (4) the ratio between the depth of the dip and the maximum value (Depth, Fig. 3j). The PSF from a single UCNP transforms from a Gaussian-like spot into a doughnut-like spot by varying the nanoparticle-mirror distance (Fig. 3a–f), which leads to four distinct curves of changes on these typical parameters (Fig. 3g–j). By establishing the calibration curves, we can precisely derive the nanoparticle-mirror distance. In particular, the areas with sharper changes in the curves indicate a better resolution for distance sensing. A cross-validation method is used to evaluate distance by taking all the four characterization curves into consideration, as shown in Supplementary Information Note 5 and Supplementary Fig. 19. The estimated localization resolutions across the different distance ranges are shown in Supplementary Fig. 20, where the sub-5 nm resolution can be achieved for the spacing ranges of 104–137, 160–186, and 282–365 nm (Supplementary Fig. 20c). Notably, the SELFI effect happens for a wide range of media, including water and air (see the simulation results in Supplementary Figs. 21 and 22), though the calibration curves are dependent on the refractive index of a media that modifies the phase of the reflected emission.

**Position-sensitive PSFs of nanoparticles' self-interference**. In Fig. 3, we illustrate how to extract quantitative features from the PSF and correlate them with the distance of the UCNP from the

**Axial distance sensing by lateral PSFs.** Figure 4 further demonstrates that the nanoscale SELFI effect provides an opportunity in widefield video-rate distance sensing towards

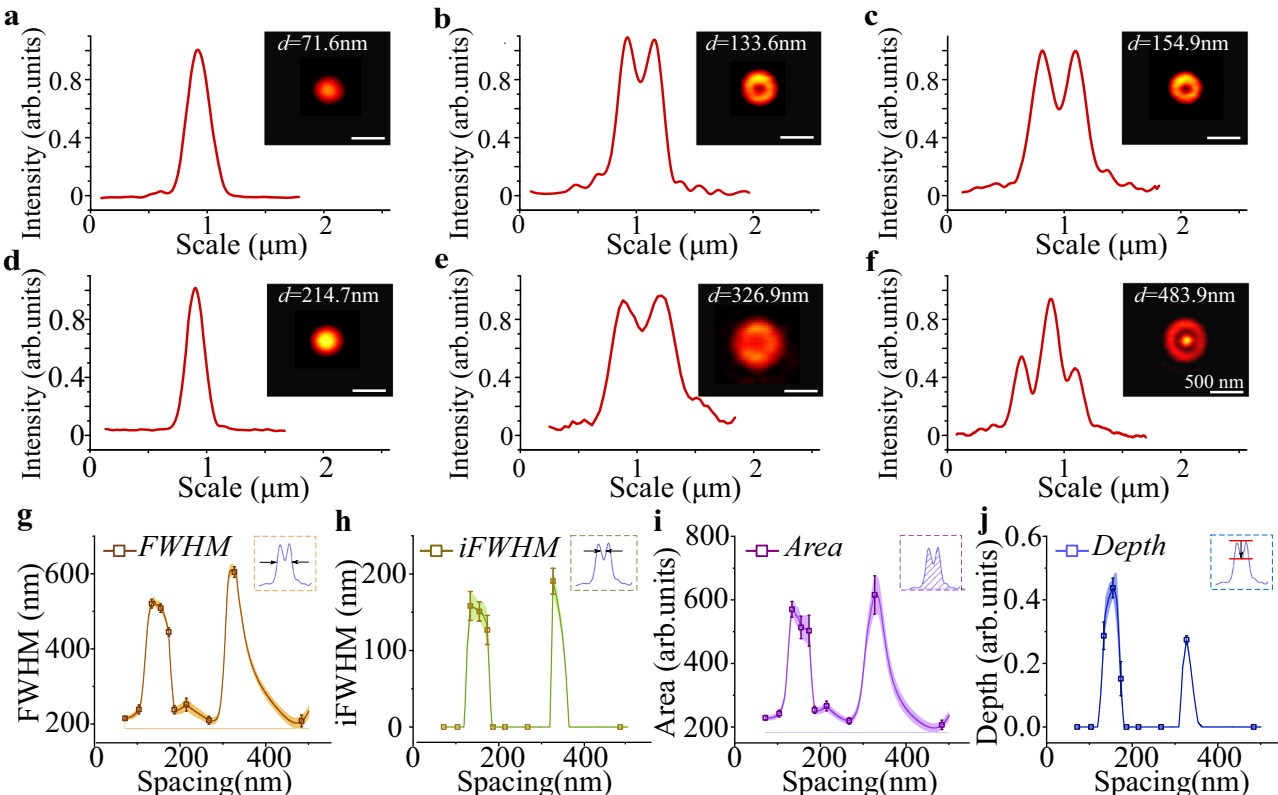

**Fig. 3 Features and quantitative analysis of the series of observed lateral PSFs used for determining the distance between UCNP and mirror. a–f**
Cross-section line profiles of the lateral PSF of UCNPs (shown as insets) for six different spacings. **g–j** The four characteristic parameters, including *FWHM*, *iFWHM*, *area*, and *depth*, respectively, were measured as a function of the distance between UCNP and mirror. Additional PSF analysis results for 455 and 800 nm emission are provided in the Supplementary Figs. 16 and 17, respectively. Further details for resolution analysis are explained in Supplementary Information Note 5. Error bars are based on the standard deviation. All scale bars in this figure are 500 nm.

sub-3 nm resolution, according to Fig. 3g–j. Both lateral and axial locations of single UCNPs can be simultaneously determined by widefield images. By dispersing UCNPs on a stair structure (Fig. 4a), the particle-mirror spacing for Steps 1, 2, and 3 are first measured by atomic force microscopy as 104.7, 121.9, and 131.9 nm, respectively (Fig. 4b). Figure 4c(i–iii) shows the contrast of typical widefield fluorescence images. Due to the SELFI effect, a slight change in the thickness, e.g., as small as 10 nm, can lead to a significant change in the lateral PSF from a Gaussian to a doughnut shape. By using the four characteristic parameters of the PSF, excellent distance sensing results are achieved, as shown in Fig. 4d–g. According to the cross-validation method (Supplementary Fig. 19), the measured values for Step 1, 2, and 3 by our fluorescent SELFI method are 103.7 nm (Fig. 4d), 124.7 nm (Fig. 4d) and 130.8 nm (Fig. 4e), respectively, suggesting that the differences between our method and the AFM values are smaller than 2.8 nm. The resolution will be further improved by using PSF pattern recognition and machine learning. Remarkably, this method only requires a single frame with exposure time as short as 20 ms used in Fig. 4c, which provides an imaging/sensing frame rate as fast as 50 Hz. This shows the superior advantage for distance sensing, compared with technologies based on the single point scanning method, including the lifetime and spectrum-based methods[8,37] that require the 2D sensing frame rate of 0.0046 Hz for a field of 50 μm by 50 μm. We further verify the resolving power of the method by measuring the heights of two batches of nanoparticles on the same substrate. As shown in Supplementary Fig. 23, the height of each particle could be resolved, and the measured average diameter difference between the two batches is 8 ± 4.8 nm which is close to the difference of

8.9 ± 2.8 nm verified by transmission electron microscope measurement (Supplementary Fig. 24).

Our method is immune to the detection defocusing (drifting of the stage), as the lateral PSF maintains its qualitative feature even with a large defocusing range, e.g., ±100 nm away from the centroid only leads to <5% variation on feature values (see Supplementary Note 7 and Supplementary Fig. 25). In Fig. 4h, we demonstrate a drifting immunized video-rate super-resolution 3D tracking of a UCNP in glycerol solution. During the long tracking period of 13 s, the cumulative displacement (Fig. 4i) of the nanoparticle shows a mono-phase diffusion, and the mean-square displacement (Fig. 4j) indicates a typical Brownian motion with a diffusion coefficient of $D = 0.0174\ \mu m^2/s$ and diffusive exponent α of 0.991. The calculated viscosity based on the diffusion coefficient is 347.7 cP, matching well with the calibrated viscosity value of glycerol (93% weight)[38]. Compared with other PSF-based single-particle tracking methods, e.g., SELFI[10] and 3dSTORM[16], our method is immune to defocusing, and therefore does not require locking the visualizing plane to the origin focus plane. To experimentally demonstrate this advantage, we periodically moved the stage from 0 to 200 nm with the increment of 20 nm per frame to artificially induce large defocus drifts, and the accurate axial locations can still be extracted as shown in Fig. 4h. Hence our method has the potential for tracking fast motions of biomolecules in living cell or organ-on-a-chip environment.

## Discussion
In summary, we demonstrated that upconversion emission SELFI from single highly doped UCNPs creates a series of unique PSF patterns that are highly sensitive to the distance between the

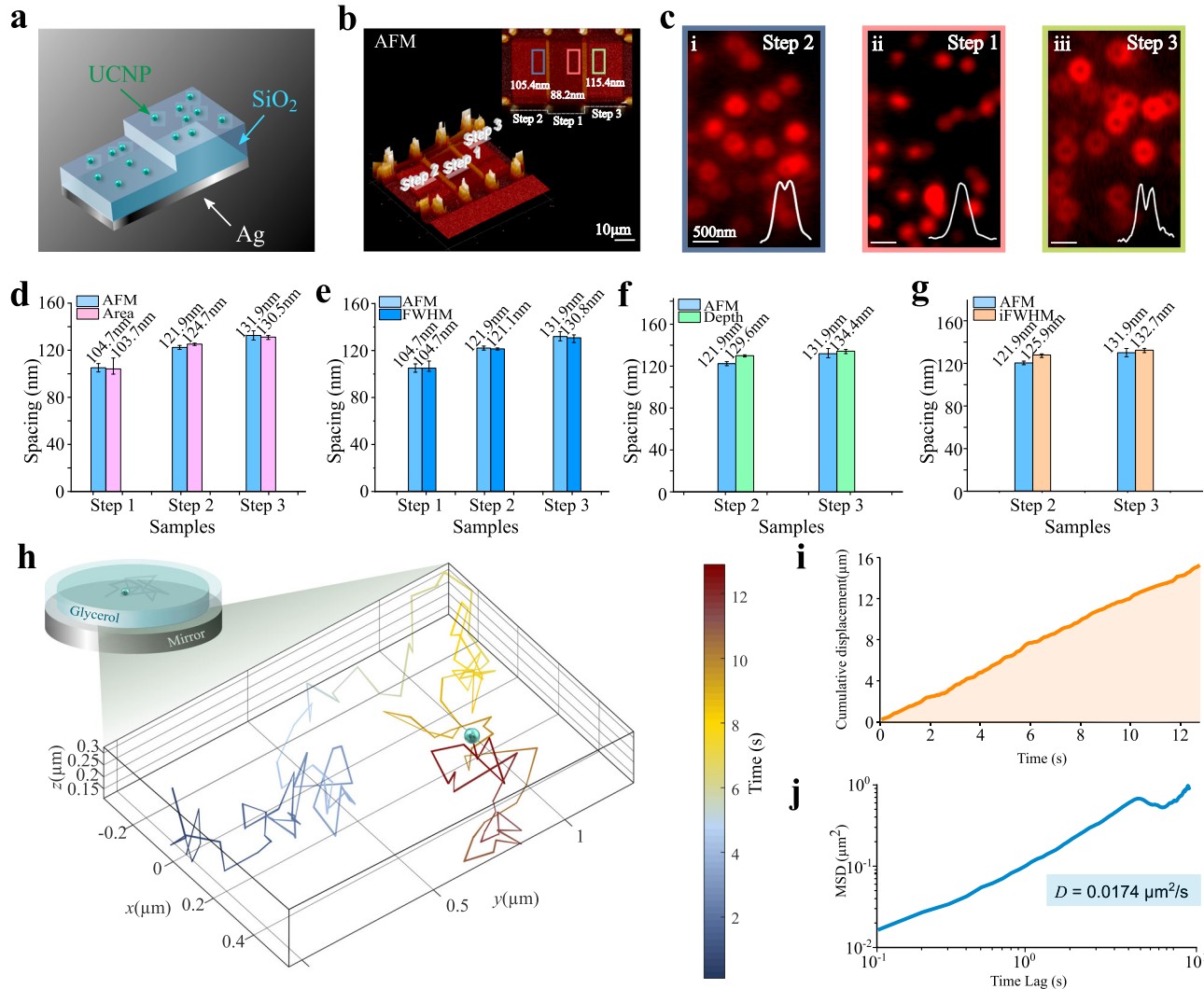

**Fig. 4 Widefield video-rate distance sensing using lateral PSFs of upconversion emission self-interference. a** The schematic shows the single UCNPs on a fabricated step structure using different thickness of SiO$_2$ on an Ag mirror substrate. **b** Atomic force microscopy (AFM) images show the measured heights of the three steps as 88.2 ± 0.8 nm, 105.4 ± 1.7 nm, and 115.4 ± 1.8 nm, respectively. The heights are measured by averaging the multiple positions within the region. **c** Widefield images of UCNPs on Step 1(ii), Step 2(i), and Step 3(iii) within the colored square area in **b**. A defocused 980 nm laser creates a widefield excitation, and the fluorescence image is collected by a CMOS camera through a blue color filter (475 ± 12.5 nm). **d**–**g** The distance sensing results by measuring the four typical characteristic parameters of the lateral PSFs. The imaging/sensing frame rate is 50 Hz. The spacings in **d**–**g** are the sum of step thickness with UCNPs' radius (16.5 nm). Hence the AFM characterized spacing for Step 1, 2, and 3 are 104.7, 121.9, and 131.9 nm, respectively. **h** 3D trajectories of a UCNP in glycerol solution, observed for 13 s (see Supplementary Movie 1). The refractive index of the glycerol solution is 1.48. The color shows the observation time. The imaging/sensing frame rate is 10 Hz for compensating the reduced intensity from the UCNP by glycerol solution. **i** Cumulative displacement and **j** mean-square displacement (MSD) analysis of the UCNPs. The time lag is the time required for the particle to be displaced by a certain amount through diffusion. The fabrication, preparation, and composition of the samples are described in Supplementary Information Note 2. The method to calculate viscosity is shown in Supplementary Information Note 6. Error bars are based on standard deviation.

emitter and a reflecting surface. By densely packing multiple emitters into a single nanoparticle, narrowband emissions with strong signal strength can be reached, so that far-field imaging and direct recognition of lateral PSFs becomes feasible. The lateral PSFs can be used to decode the axial position information for axial localization and tracking applications. Our preliminary results suggest a fast (50 Hz frame rate) widefield lateral PSF imaging can lead to axial localization sensing technology with an accuracy of 2.8 nm, verified by the AFM characterization values. Its detection speed is much faster than other SELFI-based methods based on mapping of the fluorescence spectrum[39] or the lifetime[40]. Our tracking experiment also proves its robustness to the excitation focus drifting. The attainable working range of our

method is up to 500 nm, as longer than 500 nm away from the mirror the intensity of the reflected emission from a single nanoparticle becomes too weak for SELFI. On the other hand, when the nanoparticle is too close to the mirror, e.g., smaller than 30 nm, the emission will subject to a quenching effect by the noble metal surface, which decreases the signal to noise ratio and thereby decreases the localization resolution. By taking the benefits of UCNPs, e.g., high photo-stability, multi-modalities in temperature sensing[41], and PH sensing[42], the technique demonstrated in this work can be used for multimodality single-particle tracking. The SELFI effect from a single nanoparticle may also provide a way to distinguish the orientation of a single dipole in real-time[43].

## Methods
Details and any associated references are provided in Supplementary Information.

## Data availability
All the relevant data are available from the correspondence authors upon reasonable request. Source data are provided with this paper.

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

## Acknowledgements
This work was supported by the Australian Research Council Discovery Project (DP190101058—F.W.), Australia-China Joint Research Centre for Point-of-Care Testing (ACSRF65827, SQ2017YFGH001190), Science and Technology Innovation Commission of Shenzhen (KQTD20170810110913065), National Natural Science Foundation of China (NSFC, 61729501, 51720105015), the Australian Research Council DECRA Fellowship (DE200100074—F.W.), and China Scholarship Council (Y.L.: No. 201607950010). This work was performed in part at the NSW Node of the Australian National Fabrication Facility. We thank A/Prof. Aleksandra Radenovic, Dr. Mathieu L Juan, Dr. Arti Agrawal, Mr. Niko Nikolay for discussion. O.B. acknowledges the UTS Distinguished Visiting Scholars scheme. O.B. and G.K. acknowledge funding by the Deutsche Forschungsgemeinschaft (DFG)—project number 182087777—SFB 951 (project B2, B18, and A12). S.B. acknowledges funding by DFG under Germany's Excellence Strategy—The Berlin Mathematics Research Center MATH+ (EXC-2046/1, project ID: 390685689).

## Author contributions
Z.Z. first observed the mirror-engineered emission PSFs of single upconversion nanoparticles. F.W., D.J., and Y.L. designed experiments. Y.L., Z.Z., F.W., J.Y., and X.Y. conducted the optical setup and performed the optical experiments. G.K., F.W., Y.L., S.B., O.B., and P.X. built the theoretical model. S.W. synthesized the upconversion nanoparticles. M.E.W., Z.Z., and Y.L. prepared the mirror samples. Y.L., F.W., and D.J. analyzed the results, prepared the figures, and wrote the manuscript. All authors participated in the editing of the manuscript. F.W., D.J., and O.B. supervised this project.

## Competing interests
The authors declare no competing interests.
