## [Peer Review File · Nature Communications]

REVIEWER COMMENTS

Reviewer #1 (Remarks to the Author):

In this manuscript, Liu et al. investigates the self-interference effect from a single nanoparticle. They further use the self-interference effect to designed an ultra-sensitive and fast distance sensing method. Authors claim that the multiple emission centre with random orientations can generate a series of unique patterns by a self-interference effect. These patterns can transfer the axial-location information into far-field image pattern, enabling a high- resolution z-axis distance sensing technology with high imaging speed. The experimental results and the assumption with simulation are quite comprehensive and well presented. Optical interference and its derivative applications are critical to optical physics. But the emission self-interference form a nanoscale emitter is rarely reported. The physical phenomena involved in the simulation fill this gap. And 3D tracking with extra high axial localization resolution provides a very broad prospect in bioapplication. The manuscript shows convincing results in engineering, simulation, and application. I think the novelty of the presented concept and the manuscript quality meets the publishing requirements. But, there are still some doubtful points that the authors should address before the it becomes eventually acceptable for publication.

1) The author mentions the optical setup is a conventional wild-field microscope, is there any difference if you use the laser-focused confocal based system as the imaging system? Whether the polarized excitation will affect the emission pattern?

2) In their experiments, the emission collection by a high NA(1.4) objective lens with oil immersion. Could the authors discuss the important role of NA in this technology?

3) Upconversion nanoparticle generates high brightness emission with multi-colours, is the effect showing for other wavelengths such as 650 nm or 550 nm?

4) The interference should have a significant effect on the single-particle intensity with different spacers. The author needs to provide data about how the emission intensity from single emitter changes with different spacers?

5) Figure 3g-j and Figure S10, it can be easily find that there are multiple possible solutions to the unknown distance corresponding to one given value of FWHM, iFWHM, Area, or Depth. Can these four paramerts work together to help determine the only solution to the distance?

6) In line 48, the author mentions that fluorescence self-interference is an essential method for long-distance axial localization of fluorescent molecules. The spacing distance in experiment or simulation is actually less than 500 nm, about 0.5λ . Also, what is the attainable shortest sensing distance. Figure 3g-j, the sensing resolution is variant for different distance ranges. The author should add some discussions or comments about this issues in the manuscript.

7) Line 118-120 (figure 1e-f), the authors use a fluorescence spectrum to confirm the occurrence of pronounced interference effects. Can the authors explain more about the enhancement around 700 nm?

8) Line 166-167, the authors mentioned not only the narrow emission bandwidth is crucial for self-interference, but also the particle size is one of the essential conditions, so can authors give more explanation and experiment results to demonstrate their opinion. Beyond upconversion nanoparticles, any other fluorescent nanoparticles like quantum dots and perovskite nanoparticles can also applied to this thenicque?

9) All the supplementary information should be referred to in the manuscript, but they didn't mention some of the data. The author should check it thoroughly.

10) SM, Figure S11 misses the subtitle of 'a,b,c'. In this section, figure S11b shows us the selection rules about the resolution calculation. Please give more explanation about resolution chosen around the inflection point of the feature curves.

11) Figure 1.f, there has some mismatch between the axial and unit numbers. The authors need to reorganize this figure. Line 188, according to the Figure S11c, the resolution of below 0.75 nm is more likely to be achieved from about 190 nm to 205 nm?

12) The language needs to be improved. There are many languages, e.g. Some description should be modified. Line 25, Using 'Achery target shapes' instead of 'target shapes' should be better. Line34, 'interfere' should change to 'interfere with', at the beginning of line 113 should remove the blank space. From line 180 to 187, the figure citation wrong. The 'figure 4g', 'figure4h', 'figure4i', 'figure4j', 'figure4a-f', 'figure4g-j', should be 'figure 3xx.....' line 223, the word 'de-focusing' you

used is different with line 223 'defocusing', you have used this word several times, keep them have a unify spelling. The same problem in SM, line 42, line 44, 'home made' is different with 'home-made'. SM line 171, 'AREA' should be 'Area'. 'DEPTH' should be 'Depth'.

Reviewer #2 (Remarks to the Author):

Liu et al, demonstrated the robust implementation of axial localization of sub-diffraction limited of individual upconversion nanoparticles. This method relies on the detection of the emitted electric field through self-interference in common-path interferometry geometry, allowing for highly stable coherent phase detection. The authors demonstrated localization sub-nm accuracy. Sub-nm localization of upconversion nanoparticles will open new possibilities for single nanoparticle detection and tracking.

Overall the authors clearly and nicely presented the study and provided convincing results supported with theoretical simulations. The manuscript is well written with a clear logical flow.

I'd recommend publication in Nature Communications after the following questions and comments are addressed.

1. Novotny et al. Principle of Nanooptics, explains in detail the dipole emission at any orientation near the substrate surface considering self-interference and the layer thickness. Please cite this related book.
2. Fig. 1a, authors plotted dipole emitter images for different orientations for ~ 160 nm SiO₂ thickness. They discuss that horizontal (x-y) dipoles exhibit local minima at the $z=0$. At ~ 76 nm or ~ 215 nm thicknesses, horizontal dipoles constructively interfere whereas vertical (z) dipole destructively interferes. I think they change the objective focus position to achieve maximum intensity at other thicknesses in both simulations and experiments (e.g., Fig 2.a). Please comment on that in the manuscript.
3. The interference between the direct emission field and the reflected emission field for dipoles always occurs on any substrate interfaces. For instance, direct emitted and reflected light from glass substrate interfere although very small due to very small refractive index difference between glass and immersion media
Therefore, the use of the term "without self-interference" in Line 106 without mirror could be confusing for the reader. I think the authors do not sandwich particles between two cover glasses for the without mirror comparison but I suggest to authors to clarify placing the mirror substrate.
4. Related to comment 2, please also discuss the different immersion medium's effect on the self-interference. The reflection and phase shifts will be modified due to refractive index differences. It will be valuable to have this discussion for those whom would implement this technique with different immersion e.g water and air.
5. Discussion on Gouy phase shift included in the optical phase shift (starting from line 147) requires citation from the literature. It is not clear why such small nanoparticles (much smaller than the wavelength) goes under Gouy phase shift. Hwang et.al. "Interferometry of a single nanoparticle using the Gouy phase of a focused laser beam" Optics Communication 2007, explains the Gouy phase shift for nanoscale scattering particles and suggest that they experience constant $\pi/2$ phase shift due to their size much smaller than the wavelength. Also, the explanation of the reflected phase shift (Φ_{refl}) in eq. 1 is confusing. Do the authors mean that the phase shift due to the reflection? Please revise the optical phase difference discussion.
6. In line 166, the authors pointed out that narrowband emission is critical for self-interference. The narrowband emission spectrum is the essence of this technique due to the coherence length

requirement for interference ($\sim d < \lambda / (\Delta\lambda^2)$). Please discuss the effect of the emission spectrum on the spacing requirements for self-interference detection. Also, please comment on the coherence length of the UCNP.

7. Small comments, in line 106, there is an inconsistency between text and figure hyperlinks labels -- y-dipole is Fig 1-iii. In Fig. 4d top left, there is a typo in the step number.

Reviewer #3 (Remarks to the Author):

In this manuscript, Liu et al introduce a method to axially localize and track photoluminescent point emitters or nanoparticles, through the use of a mirror substrate and the dependence of the lateral point spread function of the optical microscope on the distance of the emitter from the mirror. Based on experimental measurements and numerical simulations of the image of such emitters at various well-controlled distances from a reflective interface, the authors claim that axial localization of the emitters at an accuracy of 0.9 nm is possible using their technique.

The subject of this work is very interesting and undoubtedly has direct practical applications in optical microscopy/nanoscopy in lifesciences. The manuscript is well written.

However, to my opinion, the impact of this work is much weakened by (1) the lack of sufficient references to previous works related to the author's technique; (2) some inconsistencies in the theoretical part (both in the model and the comparison between the simulated and experimental data); and (3) the fact that the methodology used to demonstrate the main claim of the manuscript (i.e., axial localization of emitters with 0.9 nm accuracy) is questionable, which suggests that the performance of the authors' technique may be strongly overstated. Below, I explain in more detail the three points mentioned above. Based on what is written below, I recommend that major revisions of the manuscript are required before it can be considered again for publication in Nature Communications.

(1) The introduction needs to be improved in order to give a fairer overview of the state of the art. For instance, no mention of the axial resolution of TIRF, SAF, STED microscopy may be found, nor of that of STORM, PALM, or various previously reported axial localization techniques based on PSF engineering. Also, techniques where the axial position is encoded into the time domain are not limited to MIET and gMIET; see, e.g., Ref. [Jouchet et al, Biophys. J. 118, 3, 149a (2020)], among others. Moreover, the use of a mirror to improve the axial resolution in optical microscopy has been reported in several publications years before its implementation in nanoscopy as mentioned (Ref. [3]) by the authors in the first paragraph. See, e.g., in Ref. [Mudry et al, Phys Rev Lett 105, 203903 (2010)] and the many articles where the paper by Mudry et al is cited. Another example is the absence of any references to previous works on the nanoscale axial localization of nanoparticles in front of a mirror. See, e.g., in Ref. [Mock, Nano Lett. 8, 8, 2245 (2008)], where basically the same technique as introduced by the authors is used to estimate with sub-nm accuracy the thickness of a spacer layer between nanoparticles and a mirror substrate.

(2) There are some inconsistencies in the theoretical approach and the comparison between the simulated and experimental data, that the authors should fix in the revised version of their manuscript. Firstly, the authors use a set of three (colocalized?) point emitters to simulate the emission of a luminescent particle that has a diameter of 33 nm. This may be correct only at large distances from the mirror compared to the size of the particle. Why not integrating over the volume of the nanoparticle? Secondly, the distance-dependent contributions of the emitting dipoles oriented along x, y, and z may be treated independently of the distance-dependent x, y, and z-components of the excitation field only if the dipolar emitters undergo random rotational motion fast enough compared to the excitation state lifetime. Only in that case, one may consider that the emission is from one third of px, one third of py, and one third of pz-oriented electric dipoles. However, the authors do not specify whether the emitters within the nanoparticles undergo such diffusion or whether they are static. Thirdly, if the emitters inside the nanoparticles do not have a radiative quantum yield of 100%, then the px, py and pz contributions within the model should

also take into account the distance-dependent variation of the apparent quantum yield in presence of the mirror. This does not seem to be the case here. Fourthly, the authors state on lines 120 to 122 that "the results from lifetime measurements (Figure S4a) and the analytic calculations of density of states (Figure S4b) confirms the plasmonic coupling is negligible with the working range larger than 75 nm". However, such statements are not supported by Figures S4a and S4b. On the contrary, Fig. S4b shows that the LDOS parallel to the mirror decreases by not less than 40% when the particle-to-mirror distance varies from 130 to 280 nm; same observation for the LDOS orthogonal to the mirror, comparing distances of 80 and 200 nm. In addition, the lifetime measurements shown in Fig. S4a are averaged over x , y , and z -orientations of the dipoles and, thus, they do not show anything on the lifetime variations for each of the dipole orientations. From theory, non-negligible variations are expected versus the distance to the metallic surface for the in-plane and out-of-plane dipole orientations; yet, since these variations do not follow the same trend and more or less compensate each other, they are not visible in the experimental lifetime measurements, where the contributions from all dipole orientations are averaged. Fifthly, Figure 2 reveals significant discrepancies between the simulations and the experimental data, both in the xy and yz cuts, which the authors do not comment. In particular, the z -distribution of the intensity and the radii of the rings in the doughnut-shaped spots in the xy cuts do not match the model. The authors state on lines 160-161 that there is an "excellent agreement" between theory and experiment, which is not satisfying.

(3) The methodology used to demonstrate the main claim of the manuscript (i.e., axial localization of emitters with 0.9 nm accuracy) is questionable. Indeed, it is based on two reasonings that may be easily contradicted. Firstly, Figure S10 shows that the authors have estimated the axial resolution of their axial localization technique from an extrapolation of the error bars on a few experimental points, by simply linking the points with a straight segment of line, which is not accurate. Indeed, fitting the experimental data with a model would be more accurate. In addition, the resulting estimation is much dependent on the definition of the measurement errors or fit uncertainties, whatever the way such error bars are obtained, which is arbitrary. Secondly, the authors infer the axial resolution of their axial localization technique from a comparison to the thickness measurements of the spacing layer between the nanoparticles and the mirror obtained using an atomic force microscope. However, Figure 4 clearly suggests that the authors have averaged the measurements over a number of nanoparticles, which may strongly compensate for the error on each measurement and the spatial dispersion of the layer thickness due to the layer deposition technique and the inherent roughness of the metallic film, which may be higher than the claimed resolution of the authors' technique. The latter demonstration shows that the authors' technique may be useful for estimations of the thickness of a layer with sub-nm accuracy, based on measurements on several nanoparticles deposited on the layer; however, this does not conclusively demonstrate that sub-nm resolution is attained for one individual nanoparticle or emitter.

Point by point response to the reviewers' comments:

Reviewer #1 (Comments for the Author):

In this manuscript, Liu et al. investigates the self-interference effect from a single nanoparticle. They further use the self-interference effect to designed an ultra-sensitive and fast distance sensing method. Authors claim that the multiple emission centre with random orientations can generate a series of unique patterns by a self-interference effect. These patterns can transfer the axial-location information into far-field image pattern, enabling a high- resolution z-axis distance sensing technology with high imaging speed. The experimental results and the assumption with simulation are quite comprehensive and well presented. Optical interference and its derivative applications are critical to optical physics. But the emission self-interference form a nanoscale emitter is rarely reported. The physical phenomena involved in the simulation fill this gap. And 3D tracking with extra high axial localization resolution provides a very broad prospect in bioapplication. The manuscript shows convincing results in engineering, simulation, and application. I think the novelty of the presented concept and the manuscript quality meets the publishing requirements. But, there are still some doubtful points that the authors should address before the it becomes eventually acceptable for publication.

We thank the reviewer for the constructive comments and the appreciation of our work. We address the reviewer's comments and questions in the following.

1) The author mentions the optical setup is a conventional wild-field microscope, is there any difference if you use the laser-focused confocal based system as the imaging system? Whether the polarized excitation will affect the emission pattern?

The PSF modified by self-interference can only be seen with a wild-field microscope. The imaging PSF in a confocal scanning system is mainly representing the excitation PSF and detection PSF (confocal pinhole), rather than the emitting PSF, which eliminates the features at the emission distribution by self-interference. The polarized excitation will not affect the emission pattern. During the self-interference process, the nanoparticle will absorb the excitation photon and transfer the energy for emission, and the emitted light will proceed with the interference process. Hence the excitation beam does not participate the interference, and it will not affect the pattern shape. We have added “ **As the confocal microscopy confines the excitation PSF and the detection PSF through a pinhole, rather than the wide-field emitting PSF, here we employ the simple wide-field microscope (see Figure S1) with the 980 nm excitation to image the spatial distribution of the self-interference induced PSF.**” in the main text to clarify the system requirement.

2) In their experiments, the emission collection by a high NA (1.4) objective lens with oil immersion. Could the authors discuss the important role of NA in this technology?

To address the reviewer's concern, we have simulated the far-field distribution (Fourier domain image) of a UCNP with 154.9nm silica layer. As shown in the revised Figure S2, the self-interference process will modify the far-field distribution. A smaller NA will decrease the collecting angle and the resultant viewing range in the Fourier domain, which modulates the collected amplitudes of emission for dipoles along each direction (x, y, and z). More specifically, a lower NA will more strongly decrease emission from the z dipole than that from the x, and y dipoles. When the effective NA decreases to below 0.6, the amplitude from a z dipole becomes too low to be comparable with the amplitudes for x and y dipoles. Therefore, the method is

valid for optical systems with effective $NA > 0.6$. Besides, the calibration process (see Figure 3) is required for different NA, as the ratio between emission components is changed.

We have added a paragraph and the Figure S2 into the Supplementary information Note 1.

“Figure S1 is the experimental system, where a 980 nm laser is focused on the back aperture of the objective lens to generate a wide-field excitation. A CCD camera is used to record the fluorescence image for distance sensing. The objective lens is an oil-immersed lens with $NA=1.4$. The method is valid for an objective lens with other NA values. Figure S2 shows the simulated far-field distribution (or Fourier domain) from a UCNP with 154.9nm silica layer, the collection ability of the system is labelled by a black dashed line. A smaller NA will decrease the collecting angle and the resultant collecting range in the Fourier domain, which modulates the collected amplitudes of emission for dipoles along each direction (x, y, and z). Therefore, a calibration process (e.g. in Figure 3) is required for other NA values. When the effective NA decreases to below 0.6, the amplitude from a z dipole became too low to be comparable with amplitudes for x and y dipoles. Therefore, the method is valid for optical systems with effective $NA > 0.6$.”

Figure S2. Far-field distribution (Fourier domain) intensities for single (a) x-dipole, (b) z-dipole and (c) y-dipole. The dashed line indicates the collection ability of a system with an effective $NA=1.01$ (estimated from the PSF of the system).

3) Upconversion nanoparticle generates high brightness emission with multi-colours, is the effect showing for other wavelengths such as 650 nm or 550 nm?

The self-interference effect happens for all emitting wavelengths. To address the reviewer’s concern, we have conducted an extra experiment to measure the emission pattern for 650 nm, which is shown at the revised Figure S14 where a clear doughnut PSF is shown at 267.5 nm, indicating the self-interference effect.

Figure S14. The experimentally measured PSFs of the upconversion emission self-interference for UCNPs on a mirror substrate with the distance of 103.8 nm, 133.6 nm, 173.5 nm, 214.7 nm, 267.5 nm, and 326.9 nm away from the mirror surface, The radius of the UCNPs was 16.5 nm, and the emission wavelength used for detection is at 650 nm. Scale bar is 500nm.

4) The interference should have a significant effect on the single-particle intensity with different spacers. The author needs to provide data about how the emission intensity from single emitter changes with different spacers?

To adopt the reviewer’s suggestion, we have added the revised Figure S15 to show the emission intensity change for both 455 nm and 800 nm. The emission from UCNP on a mirror is generally stronger than a UCNP on a glass substrate, as the mirror will reflect back otherwise undetected emission in a forward direction. The intensity is modulated with spacing distance due to the interference effect. According to Figure S15, lower emission intensity is observed when at destructive interference, e.g. at 154.9 nm and 326.9 nm for 455nm and at 326.9nm for 800nm. Destructive interference modulates the emitting angle in such a way that part of the emitted light is beyond the collection angle of the objective lens. **We have added this information into the main text.**

“Note that the mirror will enhance the emission intensity by reflecting back the otherwise undetected emission in a forward direction. However, the enhancement depends on the spacing distance, as the destructive interference may redirect more emissions beyond the maximum collecting angle given by the NA of the optical system (see Figure S15 for the detail of change for the emission intensity).”

Figure S15. Integrated collected emission intensity for different spacing values.

(5) Figure 3g-j and Figure S10, it can be easily find that there are multiple possible solutions to the unknown distance corresponding to one given value of FWHM, iFWHM, Area, or Depth. Can these four paramerts work together to help determine the only solution to the distance?

As each pattern is unique, one distance solution can be found by considering all the four parameters. We have used this method to estimate the positions in Figure 4. We have added this information and Figure S21 into the revised supplementary information Note 5.

“To estimate the spacing within an unknown range, a cross-validation method that considering all four parameters is required. Figure S21 shows an example of using the method. (1) The values of FWHM, iFWHM, Area and Depth can be directly measured from the 2D PSF (Figure S21a, inset). We denote these four values as a_0 , b_0 , c_0 , d_0 here. (2) According to the measured FWHM, multiple corresponding spacing values (in this case denoted as Z_{a1} , Z_{a2} , Z_{a3} , Z_{a4}) will match according to the calibration curve, as shown in Figure S21a. Similarly, according to the measured iFWHM, Area and Depth values, three more sets of spacing values (denoted as Z_{b_i} , Z_{c_i} , Z_{d_i} , with $i=1,2,3,4$) will be found as shown in Figure S21b, S21c and S21d, respectively. Altogether, we obtained j potential spacing values (e.g. we have $j=16$ here for four suits of data). (3) For each of potential distances, e.g. $Z_{a1}=130.2$ nm four characterizing values ($a_{a1}=a_0$, b_{a1} , c_{a1} , d_{a1}) are found from the calibration curves, as shown in Figure S21a, S21b, 21c and 21d. (4) We use $\text{Var}_j = \sum_{k=a}^{k=d} (k_0 - k_j)^2$ to rank each of four potential spacing values (e.g. $\text{Var}_{Z_{a1}} = \sum_{k=a}^{k=d} (k_0 - k_{a1})^2$). (5) The smallest Var_j then most likely heralds the real distance value.”

Figure S21. The characteristic parameters curves of the (a) FWHM, (b) iFWHM, (c) Area, and (d) Depth. The inset in (a) is a testing PSF pattern.

6) In line 48, the author mentions that fluorescence self-interference is an essential method for long-distance axial localization of fluorescent molecules. The spacing distance in experiment or simulation is actually less than 500 nm, about 0.5λ . Also, what is the attainable shortest sensing distance. Figure 3g-j, the sensing resolution is variant for different distance ranges. The author should add some discussions or comments about this issues in the manuscript.

Theoretically, the self-interference effect is attainable from 0nm to long distance, e.g. micron meters. But due to the efficiency issue and the current experimental system, the most attainable range for this work is 50-500nm. More specifically:

At distances smaller than 30nm, the emission will still have interference with its reflected beam, but the emission efficiency may drop due to the quenching effect from the gold film. The sensing resolution is below 10nm estimated from the revised resolution estimation figure.

The mechanism is also working when the distance is larger, but since the reflected intensity is reduced, the interference effect becomes harder to detect. As shown on the simulated PSF (y-z) of 455 nm from one emitter without self-interference (Figure R1), the beam intensity achieves the first minimum around 500 nm away from the focus. Hence when the distance is larger than 500nm, the interference modulation to the pattern is very small. Note that the experimental depth of field of the emission will be longer than the simulated result due to the aberration.

We have added the above information to the revised conclusion at the main text.

“ The attainable working range of our method is up to 500 nm, as longer than 500 nm away from the mirror the intensity of the reflected emission from single nanoparticle becomes too weak for self-interference. On the other hand, when the nanoparticle is too close to the mirror, e.g. smaller than 30 nm, the emission will subject to quenching effect by the noble metal surface, which decreases the signal to noise ratio and thereby decreases the localization resolution.”

Figure R1. Simulated emitter PSF at x-z plane without a mirror substrate.

7) Line 118-120 (figure 1e-f), the authors use a fluorescence spectrum to confirm the occurrence of pronounced interference effects. Can the authors explain more about the enhancement around 700 nm?

We are very thankful to the reviewer and have double-checked this issue. Indeed, the sharp peak around 700 nm (in the previous version) was due to an artefact in our measurement. Although the emission at 700 nm should have an enhancement, there should be no sharp peak from Tm^{3+} emission. The peak might have stemmed from the home-made spectrometer. We repeated our experiment with another spectrometer system, and we confirmed the previous extra enhancement for 700nm peak as due to the previous spectrometer system. We have revised Figure 1 accordingly, which matches our conclusion from the previous result. The 455nm emission should have less enhancement, and the 800 nm emission should have more enhancement, see the revised Figure 1.

Figure 1 | Spatial self-interference of UCNP on a mirror substrate. (a) A single UCNP is placed on a mirror substrate with a silica layer as the spacer. Calculations of self-interference of dipole emitters with different orientations on a mirror substrate were performed with a finite element Maxwell's equation solver (JCMwave) for a SiO₂ spacer layer with a spacing of 154.9 nm. (i), (ii) and (iii) show different cuts through the far-field PSFs of self-interference patterns of oriented dipoles oscillating along the x, y and z-axis, respectively. The horizontal two-dimensional cuts are for x-y planes with z=0 (dipoles' centre). The vertical cuts are for x-z planes with y=0 (dipoles' centre). The emission wavelength in simulation is 455nm. (b) The total emission field from the self-interference field of x-, y- and z-dipoles. In the simulation for (a) and (b), the origin (x=0, y=0, z=0) of the coordinate system is in the centre of the UCNP, while the Ag mirror is located 154.9 nm below the origin. Experimentally, the weights of the self-interference field of x-, y- and z-

dipoles will be modified by the effective numerical aperture of the imaging collecting system, which has been considered in the simulation. (c) Wide-field fluorescence image of UCNPs in the configuration (a) with UCNP-to-mirror distance (spacing) of 154.9 nm. (d) Wide-field fluorescence image of UCNPs on a cover glass surface. The wide-field fluorescence image of UCNPs is filtered through a blue colour filter (475 +/- 12.5 nm). A defocused 980 nm laser creates the wide-field excitation. The measured emission spectrum from a single UCNP on (e) a mirror substrate with a spacing of 154.9 nm and (f) a cover glass surface. The UCNPs used in the experiment are β -NaYF₄:18%Yb³⁺/8%Tm³⁺, with a diameter of 33 nm. The size of UCNPs is characterized by transmission electron microscopy, as shown in Figure S3. (Further details on the synthesis and characterization of UCNPs as well as fabrication of the mirror substrate are provided in the supplementary information note 1). The spectra are measured by a single-particle characterization system described in detail in ref.³¹

8) Line 166-167, the authors mentioned not only the narrow emission bandwidth is crucial for self-interference, but also the particle size is one of the essential conditions, so can authors give more explanation and experiment results to demonstrate their opinion. Beyond upconversion nanoparticles, any other fluorescent nanoparticles like quantum dots and perovskite nanoparticles can also applied to this technique?

This is an important point. A larger size will induce blur at the interference pattern. We have discussed the size effect in the main text previously: “*Not only is the narrow emission bandwidth crucial for nanoscale self-interference, the small size of UCNPs with high-density emitters is also mandatory, as larger structures would cause a blurring of the interference patterns. The average phase difference of light emitted from two emitters in a UCNP with a radius of 16.5 nm is only $\Phi_{blur} = 2\pi \frac{R}{\lambda} n \sim \pi/10$ ($\lambda = 455\text{nm}$, R is the average distance²⁴), which does not prevent the observation of self-interference.*”

Besides, the larger size of the particle leads to a longer spreading of UCNPs along the axial direction, which will blur the interference pattern as well. We have added the size induce effect into the revised supplementary information Note 3 as follows:

3. Particle size effect to PSF

In our system, the feature of light field distribution is much larger than the size of the nanoparticle; hence, the lateral distribution of emitters in 33nm will not affect the diffraction-limited emission PSF. As a contrast, the axial distribution of emitters within a nanoparticle region will modify the PSF. At the revised Figure S16, we simulated the self-interference effect by integrating 204 emitters randomly distributed over the volume of a 33nm nanoparticle. Figure S16 c and S16 d are the simulated x-y PSF for the spacing of 154.9nm with the emitter at the centre of particle and randomly distributed over the particle, respectively. The axially distributed emitters inside the particle will “blur” the PSF, as the PSF of the particle is a linear combination of individual PSFs from each emitter that have slightly different axial position. Hence, the simulated calibration curve for a particle with emitters randomly located in its volume can be simplified as the convolution of a standard calibration curve (all emitter is at the centre point of particle) with a top-hat function with width of d (diameter of the particle). A larger size particle will decrease the gradient of the calibration curve and reduce the resolution. The PSFs for spacings of 71.6 nm (Figure S16 b) and 483.9 nm (Figure S16 f) have been less affected by the random emitters, as the modulation strength to PSF depends on the gradient of the characterization curve.

The reviewer is right that the self-interference mechanism can be applied to quantum dots or other nanoparticles, if we use a narrow filter to select a narrow band of emission. The challenge for other particles is to have an emission within a narrow band comparably bright as UCNP. For the same size, UCNP have stronger emission, especially within the narrow band. Besides, the near-infrared excitation wavelength for UCNP is more biologically friendly.

9) All the supplementary information should be referred to in the manuscript, but they didn't mention some of the data. The author should check it thoroughly.

We have fixed this in the revised manuscript.

10) SM, Figure S11 misses the subtitle of 'a,b,c'. In this section, figure S11b shows us the selection rules about the resolution calculation. Please give more explanation about resolution chosen around the inflection point of the feature curves.

We have revised Figure S11 to add the subtitles. To address the reviewer's concern we have added "If the characterizing value is within the error range of an inflection point (e.g. $557 < \text{Area} < 676$), the cross-section of the error region must across both the sections with spacing larger and smaller than the inflection point, as illustrated in Figure S20c (inset)." into the SI.

Figure S20. (a) The calibration error region of the FWHM curve. The calibration error estimations of iFWHM, Area, and Depth are shown in b, c, and d, respectively. The sensing resolution is estimated, as shown in the inset.

11) Figure 1.f, there has some mismatch between the axial and unit numbers. The authors need to reorganize this figure. Line 188, according to Figure S11c, the resolution of below 0.75 nm is more likely to be achieved from about 190 nm to 205 nm?

We apologize for the mistake. The values at both the main text and the SI are correct, while we forget to update Figure S11. We have fixed this issue by update the revised Figure S22.

Figure S22. (a) Values for the localization precision of the derived spacing for four calibration curves (FWHM, Area, iFWHM, Depth). (b) The selection rule for obtaining best resolution values at the different spacing region. “1”, “2”, “3” and “4” indicate the selection of FWHM, Area, iFWHM, and Depth, respectively. (c) Resulting best values for the localization precision by using the self-interference effect.

12) The language needs to be improved. There are many languages, e.g. Some description should be modified. Line 25, Using ‘Achery target shapes’ instead of ‘target shapes’ should be better. Line34, ‘interfere’ should change to ‘interfere with’, at the beginning of line 113 should remove the blank space. From line 180 to 187, the figure citation wrong. The ‘figure 4g’, ‘figure4h’, ‘figure4i’, ‘figure4j’, ‘figure4a-f’, ‘figure4g-j’, should be ‘figure 3xx.....’ line 223, the word ‘de-focusing’ you used is different with line223‘defocusing’, you have used this word several times, keep them have a unify spelling. The same problem in SM, line 42, line 44, ‘home made’ is different with ‘home-made’. SM line 171, ‘AREA’ should be ‘Area’. ‘DEPTH’ should be ‘Depth’.

We thank the reviewer for the careful review. We have fixed above language errors.

Reviewer #2 (Comments for the Author):

Liu et al, demonstrated the robust implementation of axial localization of sub-diffraction limited of individual upconversion nanoparticles. This method relies on the detection of the emitted electric field through self-interference in common-path interferometry geometry, allowing for highly stable coherent phase detection. The authors demonstrated localization sub-nm accuracy. Sub-nm localization of upconversion nanoparticles will open new possibilities for single nanoparticle detection and tracking. Overall the authors clearly and nicely presented the study and provided convincing results supported with theoretical simulations. The manuscript is well written with a clear logical flow. I'd recommend publication in Nature Communications after the following questions and comments are addressed.

We thank the reviewer for the positive view on our manuscript and the comments, which we address in the following.

1. Novotny et al. Principle of Nanooptics, explains in detail the dipole emission at any orientation near the substrate surface considering self-interference and the layer thickness. Please cite this related book.

We appreciate the reviewer's suggestion to cite this great book. We have used the method shown in the book to calculate the density of state. Chapter 10 of the book describes the electric field distribution radiated from a dipole near a substrate. Here we are investigating the far-field point spread function for the same case. We have cited the book at the revised manuscript.

We have added “**Though the electric field distribution radiated from a dipole near a substrate is well developed³², its far-field PSF is still challenged to be expressed as a mathematic formula.**” into the main text.

2. Fig. 1a, authors plotted dipole emitter images for different orientations for ~160 nm SiO₂ thickness. They discuss that horizontal (x-y) dipoles exhibit local minima at the z=0. At ~76 nm or ~215 nm thicknesses, horizontal dipoles constructively interfere whereas vertical (z) dipole destructively interferes. I think they change the objective focus position to achieve maximum intensity at other thicknesses in both simulations and experiments (e.g., Fig 2.a). Please comment on that in the manuscript.

Firstly, we want to apologize for a typo error in the main text. The first distance should be 71.6 nm rather than 76.1 nm, we have fixed this error at the revised manuscript. According to the phase analysis of the self-interference, the interference types (both constructive and destructive) for vertical and horizontal dipoles are the same along the z-axis. Therefore, all dipoles subject to constructive interference at 154.9 nm. However, if the emission polarization is along the z-axis, they cannot interfere along z-axis. Therefore, only the horizontal dipoles have a minimum on their z-axis distribution. We have added “**The z-dipole does not have a minimum along the z-axis, as its emission does not propagate along the z-axis.**” to the main text to clarify this.

In our experimental and simulation results, including Figure 2a, Figure S5 and Figure S6, the objective focus is positioned at the particle's position. This position is found by the following procedure: (1) Using Gaussian beam focus reflection to locate the metallic surface. (2) Moving the focus away from the mirror surface with the distance equal to the AFM characterized spacing value plus half of the particle size (e.g. d in Figure 2). The emission intensity at the particle's position is not always the maximum. To address reviewer's concern,

we have added “The nanoparticles locate at the measuring focal plane where $z=0$.” at the main text, we have added “Experimentally the $z=0$ plane is determined by moving the focus above the mirror surface with spacing distance. The z position of the mirror surface is located by the reflection of a Gaussian beam.” to the caption of Figure 2.

We have found a mistake in Figure 2e, the image is not from the particle plane. During the data analysis process for $d=326.9\text{nm}$ samples, we added a z -axis offset by mistake. We have fixed this mistake and updated Figure 2e, Figure S10, and Figure S20. According to the new pattern, we have revised Figure 3e and Figure 3g-j. With the new calibration curve, we have revised the Figure 4 and Figure S22.

3. The interference between the direct emission field and the reflected emission field for dipoles always occurs on any substrate interfaces. For instance, direct emitted and reflected light from glass substrate interfere although very small due to very small refractive index difference between glass and immersion media. Therefore, the use of the term “without self-interference” in Line 106 without mirror could be confusing for the reader. I think the authors do not sandwich particles between two cover glasses for the without mirror comparison but I suggest to authors to clarify placing the mirror substrate.

Here we intended to say that the 2D PSF of dipoles with self-interference are similar to the simulated 2D PSF of dipoles from previous literatures where the dipoles are placed inside a non-reflecting medium, but the 3D PSF are totally different. To avoid confusion, we rephrased the sentence as “Two-dimensional cuts (x - y plane) at $z=0$ for dipoles oriented along the x -, y -, and z -axes are similar to the traditional PSF for dipoles without the self-interference between their emission and the reflected emission³³. However, in striking difference to the traditional emission without self-interference both x -dipole and y -dipole feature a local intensity minimum along the z -axis, as shown at the z - x plane in Figure 1a (i) and Figure 1a (iii), respectively.”.

4. Related to comment 2, please also discuss the different immersion medium's effect on the self-interference. The reflection and phase shifts will be modified due to refractive index differences. It will be valuable to have this discussion for those whom would implement this technique with different immersion e.g water and air.

The reviewer is right in principle. This self-interference is a robust effect, it works for a wide range of media including water, air and oil. The reflection efficiency will be different for different media. This reflection efficiency could be estimated by the Fresnel law with a normal incident angle, where $R = |(n_2 - n_1)/(n_2 + n_1)|^2$. The calculated reflection efficiencies of Ag mirror within mediums of water ($n_1=1.33$), air ($n_1=1$) and immersion oil ($n_1=1.5$) are 0.976, 0.98 and 0.974, respectively. Hence the change in reflection from conventional media can be ignored.

The different refractive index for media will also modify the phase of the reflected beam. A smaller value of the refractive index leads to a smaller optical path induced phase change and a larger Gouy phase change. These phase changes will modulate the strongest interference position, which modifies the characteristic curves (e.g. Figure 3g-j). Hence the calibration curve must be measured for each different medium individually. We have simulated the x - y patterns for water ($n=1.33$) and air ($n=1$), as shown at the revised **Figure S12 and S13**, respectively. These simulation results confirm the self-interference effect will happen within a wide range of media. It is notable that the refractive index of glycerol solution is 1.48, which is almost the same at the refractive index for immersion oil (1.5). Therefore, the tracking application in Figure 4 could use the calibration curves shown in Figure 3. We have added these refractive index values to the revised main text.

We also added “ Notably, the self-interference effect happens for a wide range of media, including water and air (see the simulation results in Figure S12 and S13), though the calibration curves are dependent on the refractive index of a media that modifies the phase of the reflected emission.” to the main text to clarify the effect from different media.

Figure S12. The simulated PSFs of the self-interference for a particle on a mirror substrate with spacing distances of 71.6 nm, 103.8 nm, 133.6 nm, 154.9 nm, 173.5 nm, 186.6 nm, 214.7 nm, 267.5 nm, 326.9 nm, and 483.9 nm, at the x-y plane. The surrounding medium is water with a refractive index of 1.33.

Figure S13. The simulated PSFs of the self-interference for a particle on a mirror substrate with spacing distances of 71.6 nm, 103.8 nm, 133.6 nm, 154.9 nm, 173.5 nm, 186.6 nm, 214.7 nm, 267.5 nm, 326.9 nm, and 483.9 nm, at the x-y plane. The surrounding medium is air with a refractive index of 1.

5. Discussion on Gouy phase shift included in the optical phase shift (starting from line 147) requires citation from the literature. It is not clear why such small nanoparticles (much smaller than the wavelength) goes under Gouy phase shift. Hwang et.al. “Interferometry of a single nanoparticle using the Gouy phase of a focused laser beam” *Optics Communication* 2007, explains the Gouy phase shift for nanoscale scattering particles and suggest that they experience constant $\pi/2$ phase shift due to their size much smaller than the wavelength. Also, the explanation of the reflected phase shift (Φ_{refl}) in eq. 1 is confusing. Do the authors mean that the phase shift due to the reflection? Please revise the optical phase difference discussion.

We have carefully checked the reference by Hwang *et.al.* We have found they used the same equation as we do. According to Fig. 2 and the description text before equation (6) at Hwang *et.al.*'s paper, the Gouy phase shift with a distance z away from the focus is $\Phi_G = -\tan^{-1}(\frac{z}{z_R})$, thus the Gouy phase from the focus to the positive far-field ($z=+\infty$) is $-\pi/2$. The phase accumulation from the position z to the positive far-field is $\Phi_{G,inc}(z) = -\frac{\pi}{2} + \tan^{-1}(\frac{z}{z_R})$. Here z_R is the Rayleigh range. These equations suggest that when we are looking at far-field ($z=+\infty$), the Gouy phase is a constant.

In our case, we are investigating the interference around the focus plane. Hence the Gouy phase is $\Phi_G = -\tan^{-1}(\frac{z}{z_R})$. In our equation (2) the expression is $\Phi_{Gouy}(z) = \arctan \frac{2\Delta z}{z_0}$. Here $z_0 = z_R$ that is the Rayleigh range. Δz is the distance between the UCNP and the mirror. Hence $2\Delta z$ is the travel distance z for Φ_G . The negative sign is included in equation (1). To address the reviewer's concern, we have cited the reference by Hwang *et.al.*, and we have revised the symbols in equation (2) to match with the equation in the cited reference.

$$\Delta\phi(\Delta z) = \phi_z(\Delta z) + \phi_{refl} + \phi_{Gouy}(\Delta z) \quad (1)$$

$$\phi_{Gouy}(z) = -\tan^{-1} \frac{2\Delta z}{z_R} \quad (2)$$

The ϕ_{refl} represents the external reflection induced phase change. When the beam is within medium 1 (n_1) and reflected at the interface between medium 1 and medium 2 (n_2), and $n_2 > n_1$, according to the Fresnel equation, the $\phi_{refl} = \pi$, which is a constant. To clarify this phase term, we have rewritten the description as:

“ We now provide an analytic approach to give an idea of the behavior of the PSF patterns along the z-Axis. The emission from a nanoparticle propagates to the mirror, is reflected and returns to the nanoparticle to interfere with the unreflected emission. The propagation distance induces a phase change (ϕ_z). The reflection of the emission at the dielectric interface of a low refractive index medium (SiO_2) and a high refractive index medium (Ag) induces an external phase shift $\phi_{refl} = \pi$, according to the Fresnel law.”

6. In line 166, the authors pointed out that narrowband emission is critical for self-interference. The narrowband emission spectrum is the essence of this technique due to the coherence length requirement for interference ($\sim d < \lambda / (\Delta\lambda^2)$). Please discuss the effect of the emission spectrum on the spacing requirements for self-interference detection. Also, please comment on the coherence length of the UCNP.

In the upconverting system, the emission is from the atomic transition. There is a direct relation between the coherence time τ_c , the coherence length l_c and the spectral width $\Delta\omega$. Figure S23 shows the emission spectrum from a typical UCNP. Here we assume each of dipoles inside a UCNP provides the same emission spectrum. Then the FWHM of the emission peak with angular frequency axis ($\Delta\omega$) can be measured from Figure S23b. The coherence length can be estimated by:

$$l_c = c * \tau_c / n = c / (n\Delta\omega)$$

here c is the speed of light, n is the refractive index of the medium. The calculated coherence length for 455 nm and 800 nm are 3.1 μm and 2.7 μm , respectively. In this work, our z-axis viewing range is below 500 nm, which is within the coherence length for both 455 nm and 800 nm.

For emitters with broadband emission, such as dye molecules, a narrowband filter is required to maintain the coherence length above 500 nm. For this case, the bandwidth of the filter is required to be narrow than 44 nm for a central wavelength of 455 nm, according to the below equation.

$$\Delta\omega = \frac{2\pi c \Delta\lambda}{\lambda^2} = \frac{c}{n l_c}$$

$$\Delta\lambda = \frac{\lambda^2}{2\pi n l_c}$$

We have added the above information into the revised Supplementary information as Note 7.

“**Note 7: Bandwidth requirement for the self-interference**”

To achieve efficient interference around the nanoparticle, a narrowband emission is required. In the upconverting system, the emission is from the atomic transition. There is a direct relation between the coherence time τ_c , the coherence length l_c and the spectral width $\Delta\omega$. Figure S23 shows the emission spectrum from a typical UCNP. Here we assume each of dipoles inside a UCNP provides the same emission

spectrum. Then the FWHM of the emission peak with angular frequency axis ($\Delta\omega$) can be measured from Figure S23b. The coherence length can be estimated by:

$$l_c = c * \tau_c / n = c / (n\Delta\omega) \quad (S3)$$

here c is the speed of light, n is the refractive index of the medium. The calculated coherence length for 455 nm and 800 nm are 3.1 μm and 2.7 μm , respectively. In this work, our z-axis viewing range is below 500 nm, which is within the coherence length for both 455 nm and 800 nm.

For emitters with broadband emission, such as dye molecules, a narrowband filter is required to maintain the coherence length above 500 nm. For this case, the bandwidth of the filter is required to be narrow than 44 nm for the central wavelength of 455 nm, according to below equation.

$$\Delta\omega = \frac{2\pi c \Delta\lambda}{\lambda^2} = \frac{c}{n l_c} \quad (S4)$$

$$\Delta\lambda = \frac{\lambda^2}{2\pi n l_c} \quad (S5)$$

Figure S23. The emission spectrum from a single UCNP on a glass substrate with the horizontal axis as (a) wavelength and (b) angular frequency.”

7. Small comments, in line 106, there is an inconsistency between text and figure hyperlinks labels -- y-dipole is Fig 1-iii. In Fig. 4d top left, there is a typo in the step number.

We appreciate the reviewer for the carefully checking. We have corrected at the revised manuscript.

There is no typo in Figure 4d, as we compared Step1 with Step 3 directly. For each of Step, we selected one region to test the thickness. To avoid confusion, we have reorganized the Figure 4. By putting the individual imaging areas as separated images (Figure 4c-i, 4c-ii, and 4c-iii). The revised caption and figure are shown below.

Figure 4 | Wide-field video-rate distance sensing using lateral PSFs of upconversion emission self-interference. (a) The schematic shows the single UCNP on a fabricated step structure using different thickness of SiO₂ on an Ag mirror substrate. (b) Atomic force microscopy (AFM) images show the measured heights of the three steps as 88.2±0.8 nm, 105.4±1.7 nm, and 115.4±1.8 nm, respectively. The heights are measured by averaging the multiple positions within the region. (c) Wide-field images of UCNP on Steps 1(ii), Step 2(i) and Step 3(iii) within the coloured square area in (b). A defocused 980 nm laser creates a wide-field excitation, and the fluorescence image is collected by a CMOS camera through a blue colour filter (475 ± 12.5 nm). (d)-(g) is the distance sensing results by measuring the four typical characteristic parameters of the lateral PSFs. The imaging/sensing frame rate is 50Hz. The spacings in (d)-(g) are the sum of step thickness with UCNP's radius (16.5 nm). Hence the AFM characterized spacing for Step 1, 2, and 3 are 104.7 nm, 121.9 nm, and 131.9 nm, respectively. (h) 3D trajectories of a UCNP in glycerol solution, observed for 13 seconds (see Supplementary Movies S1). The refractive index of the glycerol solution is 1.48. The colour shows the observation time. The imaging/sensing frame rate is 10Hz for compensating the reduced intensity from the UCNP by glycerol solution. (i) Cumulative displacement and (j) mean-square displacement (MSD) analysis of the UCNP. The time lag is the time required for the particle to be displaced by a certain amount through diffusion. The fabrication, preparation and composition of the samples are described in the supplementary information Note 1. The method to calculate viscosity is shown at supplementary information Note 6.

Reviewer #3 (Comments for the Author):

In this manuscript, Liu et al introduce a method to axially localize and track photoluminescent point emitters or nanoparticles, through the use of a mirror substrate and the dependence of the lateral point spread function of the optical microscope on the distance of the emitter from the mirror. Based on experimental measurements and numerical simulations of the image of such emitters at various well-controlled distances from a reflective interface, the authors claim that axial localization of the emitters at an accuracy of 0.9 nm is possible using their technique. The subject of this work is very interesting and undoubtedly has direct practical applications in optical microscopy/nanoscopy in lifesciences. The manuscript is well written. However, to my opinion, the impact of this work is much weakened by (1) the lack of sufficient references to previous works related to the author's technique; (2) some inconsistencies in the theoretical part (both in the model and the comparison between the simulated and experimental data); and (3) the fact that the methodology used to demonstrate the main claim of the manuscript (i.e., axial localization of emitters with 0.9 nm accuracy) is questionable, which suggests that the performance of the authors' technique may be strongly overstated. Below, I explain in more detail the three points mentioned above. Based on what is written below, I recommend that major revisions of the manuscript are required before it can be considered again for publication in Nature Communications.

We thank the reviewer for taking the time and carefully reading our manuscript. We took the valuable comments very seriously and have tried to address them in the following. We revised our manuscript accordingly.

(1) The introduction needs to be improved in order to give a fairer overview of the state of the art. For instance, no mention of the axial resolution of TIRF, SAF, STED microscopy may be found, nor of that of STORM, PALM, or various previously reported axial localization techniques based on PSF engineering. Also, techniques where the axial position is encoded into the time domain are not limited to MIET and gMIET; see, e.g., Ref. [Jouchet et al, Biophys. J. 118, 3, 149a (2020)], among others. Moreover, the use of a mirror to improve the axial resolution in optical microscopy has been reported in several publications years before its implementation in nanoscopy as mentioned (Ref. [3]) by the authors in the first paragraph. See, e.g., in Ref. [Mudry et al, Phys Rev Lett 105, 203903 (2010)] and the many articles where the paper by Mudry et al is cited.

To adopt the reviewer's suggestion, we have rewritten the introduction section to include the most of axial resolution sensing technologies. We have cited the references suggested by the reviewer. In the revised introduction, we summarized the approaches by confining the excitation region and emission region; approaches by encoding the axial position into time, spectrum, and phase domain information; and the approaches by PSF engineering. The revised section is shown below:

“Super-resolution localization of fluorescent emitters, particularly along the axial axis, poses a key challenge in optical microscopy. Excitation confinement based approaches are relatively straightforward. Total internal reflection fluorescence (TIRF) microscope confines the excitation within a thin region above the substrate surface to achieve an axial resolution of 100-200 nm¹. Single plane illumination microscopy (SPIM) uses the narrower lateral section of the excitation beam to achieve optical sectioning perpendicular to the axial direction, providing an axial resolution of 450-750 nm.² Axial interference is an efficient way to confine the excitation region. I⁵M microscopy uses two opposing objectives with two incoherent incident beams to interfere each other so as to achieve 100 nm axial resolution.³ Similarly, 4Pi microscopy uses the optical interference of two excitation focal spots to achieve an axial resolution of 75 nm.⁴ Placing a mirror after the focal point will generate an isotropic excitation spot, due to the interference between the excitation

and its reflection⁵, which confines the confocal PSF into a 110-nm spot.⁶ Similar to the strategy by confining the excitation region, the confinement of emission regions also enhances the axial resolution. Supercritical angle fluorescence microscopy (SAF) detects the emission above the critical angle of a medium interface, which creates an emission collection region of 100-200 nm above the interface. By applying an axial hollow depletion beam to deactivate the fluorophores near the focal point, stimulated emission depletion (STED) microscopy can achieve 150 nm axial resolution⁷.

Approaches by encoding the axial position information into the time and spectrum domains are the indirect ways, but often with higher axial localization accuracy. For example, in the time domain, metal- or graphene-induced energy transfer (MIET or gMIET) approaches rely on the lifetime measurement that is steeply dependent on distance, when fluorescent molecules are nearby a metal⁸ or graphene surface⁹. These methods provide an axial resolution from a few nanometres to the ångström level. However, the achievable localization accuracy is limited by the range of energy transfer interaction (L), following the relationship L/\sqrt{N} , where N is the budget of photons from a single molecule. For example, for MIET using gold or silver films, L ranges from ~ 150 nm to 200 nm, so that the localization accuracy for single molecules is around 5 nm to 6 nm at a photon budget of 1,000 photons; for gMIET using a graphene substrate, L is reduced to ~ 20 nm, which explains why sub-nanometre localization accuracy can be achieved. These energy transfer-based technologies require the measuring sample closing to the substrate surface (e.g. < 200 nm).

Axial position information can also be encoded in the spectral domain through emission's self-interference effect, which provides a long axial sensing range. Because spontaneous emission has low degree of coherence, external references cannot be used, and self-interference is the only way to manage the interference for the incoherent fluorescence. Spectral self-interference microscopy (SSFM) uses both the reflecting mirrors and large spacers ($> 10\lambda$) to decode the vertical position of a fluorescent layer by measuring the spectrum fringes^{10,11}, with high accuracy of 0.2 nm. Mock *et al.* locate a gold nanoparticle above a metallic substrate to selectively enable the surface plasmon polariton (SPP) and the localized surface plasmon (LSP). This, in turn, provides a distance-dependent spectrum profile, resulting in an axial resolution < 1 nm¹². While this method is restricted to metallic nanoparticle and substrate, the total internal reflectance (TIR) illumination has limited its sensing range smaller than 100 nm. Above listed methods by transferring the axial information to the spectrum or time domain typically require the sophisticated time-resolved or spectra-analysis systems and often have a limited sensing speed by 2D lifetime or spectrum mappings.

The axial position can also be encoded into the optical phase variation. Shtengel *et al.* use a multi-path emission interferometric system to retrieve the position information from the phase information of the emitting photons. Such an interferometric photoactivated localization microscopy (iPALM) provides a sub-20 nm axial resolution¹³. Most recently, Juchet *et al.* developed a modulated localization (ModLoc) strategy, adopting from the amplitude modulated LIDAR approach, and achieved a 7.5 nm axial resolution¹⁴. ModLoc is based on a unique lock-in four-imaging-channel system with the interference excitation pattern to retrieve the phase information of the emitter, which can be transferred into the axial position of the emitter. These methods require both the complicated interferometric arrangement and the phase retrieving systems.

Approaches based on emission PSF engineering can detect the axial position for a larger range. The PSF can be modulated by defocusing¹⁵, optical astigmatism¹⁶, and spatial light phase mask^{17,18}. The localization can be achieved by measuring the PSF profile change. The recently reported approach, Self-Interference microscopy (SELI) generates fluorescence self-interference within each detected PSF, so that in a single camera frame, parallel analysis of the quantitative fluorescence intensity and wavefront curvature can lead to both lateral and axial localizations of single fluorescent molecules¹⁹ and video-rate tracking of quantum dots²⁰. As these methods are too sensitive to the focus position, the focus drifting will affect the accuracy. Plasmonic coupling induced polarization selection can also modify the PSF, however, the working range is limited (e.g. < 40 nm)¹².”

Another example is the absence of any references to previous works on the nanoscale axial localization of nanoparticles in front of a mirror. See, e.g., in Ref. [Mock, Nano Lett. 8, 8, 2245 (2008)], where basically the same technique as introduced by the authors is used to estimate with sub-nm accuracy the thickness of a spacer layer between nanoparticles and a mirror substrate.

We appreciate the reviewer point out *Mock et al.*'s work, which is related to this work. We have cited this reference. The mechanism of their technology is different from ours. *Mock et al.* locate a gold nanoparticle above a metallic substrate to selectively enable the surface plasmon polariton (SPP) and the localized surface plasmon (LSP). This, in turn, provides a distance-dependent spectrum profile. The distance can be calculated from the changes in the spectrum with high resolution (<1 nm). In addition, the plasmonic coupling induced polarization selection will modify the PSF. The change in PSF can be used to sense the distance change. Limited by the working mechanism, this method is limited to metallic nanoparticles and the metallic substrate, and the total internal reflectance (TIR) illumination, with limited sensing range (e.g. < 100 nm). While our method uses self-interference to modify the PSF of a nanoparticle with all directional dipoles. Hence our method could work for a distance up to 500nm, and the reflection surface is not limited to a metallic surface. We have added a paragraph at the introduction to introduce the works using PSF engineering for distance sensing.

(2) There are some inconsistencies in the theoretical approach and the comparison between the simulated and experimental data, that the authors should fix in the revised version of their manuscript. Firstly, the authors use a set of three (colocalized?) point emitters to simulate the emission of a luminescent particle that has a diameter of 33 nm. This may be correct only at large distances from the mirror compared to the size of the particle. Why not integrating over the volume of the nanoparticle?

In the revised Figure 2, we focused on showing the general self-interference effect. A simple model that all emitters are positioned at the centre point is more suitable to capture the relevant physics which already nicely reproduces the main observations. We take this as an indication for the general applicability of this concept. We put in-depth discussion of the effects from particle size, emission wavelength, immersion media, and numerical aperture to the supplementary information. In the revised Figure 3, to produce a quantitative fitting of the experimental data, we have considered the particle size effect.

In our system, the feature of light field distribution is much larger than the size of nanoparticles, hence, the lateral distribution of emitters in 33nm will not affect the diffraction limited emission PSF. As a contrast, the axial distribution of emitters within a nanoparticle region will modify the PSF. At the revised Figure S16, we simulated the self-interference effect by integrating 204 emitters randomly distributed over the volume of a 33nm nanoparticle. Figure S16c and S16d are the simulated x-y PSF for the spacing of 154.9nm with emitters at the centre of particle and randomly distributed over the particle, respectively. The axially distributed emitter inside the particle will “blur” the PSF, as the PSF of the particle is a linear combination of individual PSFs from each emitter that has a slightly different axial position. Hence, the simulated calibration curve for a particle with emitters randomly located in its volume can be simplified as the convolution of the standard calibration curve (all emitter is at the centre point of the particle) with a top-hat function with a width of d (diameter of the particle). A larger size particle will decrease the gradient of the calibration curve and reduce the resolution. The PSFs for spacings of 71.6 nm (Figure S16b) and 483.9 nm (Figure S16f) have been less affected by the random emitters, as the modulation strength to PSF depends on the gradient of the characterization curve.

We added “ Besides, the larger size of particles will result in a wider distribution of emitters along the z-axis, and thereby decreasing the gradient of the calibration curve (see Supplementary information note 4: 3).” into the main text.

We have added the size induced effects into the revised supplementary information Note 3.

3. Particle size effect to PSF

In our system, the feature of light field distribution is much larger than the size of the nanoparticle; hence, the lateral distribution of emitters in 33nm will not affect the diffraction-limited emission PSF. As a contrast, the axial distribution of emitters within a nanoparticle region will modify the PSF. At the revised Figure S16, we simulated the self-interference effect by integrating 204 emitters randomly distributed over the volume of a 33nm nanoparticle. Figure S16 c and S16 d are the simulated x-y PSF for the spacing of 154.9nm with the emitter at the centre of particle and randomly distributed over the particle, respectively. The axial distributed emitters inside the particle will “blur” the PSF, as the PSF of the particle is a linear combination of individual PSFs from each emitter that have slightly different axial position. Hence, the simulated calibration curve for a particle with emitters randomly located in its volume can be simplified as the convolution of a standard calibration curve (all emitter is at the centre point of particle) with a top-hat function with width of d (diameter of the particle). A larger size particle will decrease the gradient of the calibration curve and reduce the resolution. The PSFs for spacings of 71.6 nm (Figure S16 b) and 483.9 nm (Figure S16 f) have been less affected by the random emitters, as the modulation strength to PSF depends on the gradient of the characterization curve.

Figure S16. The simulated x-y PSFs for UCNPs with spacings of 71.6 nm (a, b), 154.9 nm (c, d) and 483.9 nm (e, f). Three emitters are located at the centre of a nanoparticle. (a, c, e). 204 emitters are located randomly inside the nanoparticle (b, d, f).

Secondly, the distance-dependent contributions of the emitting dipoles oriented along x, y, and z may be treated independently of the distance-dependent x, y, and z-components of the excitation field only if the dipolar emitters undergo random rotational motion fast enough compared to the excitation state lifetime. Only in that case, one may consider that the emission is from one third of p_x , one third of p_y , and one third of p_z -oriented electric dipoles. However, the authors do not specify whether the emitters within the nanoparticles undergo such diffusion or whether they are static.

The UCNPs have a unique upconverting energy transfer system, where the sensitizer ions will absorb the excitation photons, then transfer the energy to different energy levels at the emitter ions for emission. Hence

the emission from emitting dipoles is independent of the excitation field. We discussed more the energy transfer mechanism at the revised Note 3 in Supplementary information, as a reply to the next comment. We have added “**In the typical process of photon upconversion, sensitizer ions (e.g. Yb³⁺) absorb the lower level energy excitation photons at near infrared, and transfer the sensitized energy to their nearby emitter ions (e.g. Tm³⁺). It is the emitter ion with metastable intermediate states that can nonlinearly upconvert the energy onto the higher levels and eventually emit the anti-Stokes luminescence. In such a nanoscale system with a network of at least thousands of sensitizers and hundreds of emitters, the orientation of emitting dipole is independent of the excitation field.**” at the main text to clarify the emission is independent with excitation.

The emission is statistically split to x, y and z with equal amplitudes, as the emitting dipoles inside UCNPs do not have a preferential direction. We simplify the case as estimating the occurrence of all dipoles in different directions. Assuming a particle has n dipoles and the dipole can only be aligned along with three directions (x, y and z). Then according to Borel’s law of large numbers, the occurred case along one direction over total case number is $N_n/n \rightarrow 1/3$. When dipole can be in any direction, the case is almost the same as the case that the dipole can only along with three directions. A dipole can be decomposed as many small components along only x, y and z direction, and total case number along one direction would be 1/3. In our nanoparticle system, we have more than 3000 emitters in a particle which enables us to use Borel’s law of large numbers. Then we assume the dipole components along each of the directions are the same. We used three dipoles to represents the many emitters. We have added “**According to the Borel’s law of large numbers, the emitters are randomly oriented to provide emissions from all three orthogonal dipole orientation components with statistically equal amplitude.**” to the main text to clarify the information.

Thirdly, if the emitters inside the nanoparticles do not have a radiative quantum yield of 100%, then the p_x , p_y and p_z contributions within the model should also take into account the distance-dependent variation of the apparent quantum yield in presence of the mirror. This does not seem to be the case here.

We appreciate the reviewer’s question. In our previous simulation, we have set the quantum yield as 100%, then the change of the local density of state (LDOS) could represent the emission enhancement ratio. We have embedded the emission enhancement ratio in our previous simulation. However, as the reviewer mentioned, the quantum yield for upconversion system is not 100%, and the energy transferring system is very different from the systems for single dipoles and semiconductors.

To address the reviewer’s concern, we have modified our modelling by taking the distance-dependent variation of quantum yield and the local density of the state. We also include the up-converting energy transfer system into modelling for more accurate results. The resultant emission enhancement ratio has a variation between 0.94 and 1.05. The detail is shown below.

We have added a section of “**1. Distance induced energy transfer variation**” into the revised supplementary information Note 3 to show the detail of estimating the emission enhancement ratio. *Dawei Lu et al.* (ACS Nano 2014, 8, 8, 7780–7792) have developed the plasmon enhancement mechanism for Yb³⁺ and Er³⁺-co-doped upconverting system. Here we further extend the mechanism for Yb³⁺ and high Tm³⁺ co-doped system. **We have used the calculated emission enhancement ratio as the new weighing amplitude during our simulation.** We have updated the simulation result in **Figure 2A-2L, Figure S5A-5T, Figure S6A-6T**. The added section is shown below:

1. Distance induced energy transfer variation

The plasmon coupling between a UCNP and a metallic film is more complex than the coupling between a dipole and a metallic film, as a UCNP has a multi-level energy upconverting system. Figure S5 is the diagram of the up-converting energy transfer process for Yb³⁺ and Tm³⁺ co-doped system. To simplify the energy transferring process, we use a five-energy level model for the simulation, where n5, n4 and n1 stand for energy levels of ¹D₂, ¹G₄ and ³H₆ respectively. The n3 is the degeneration of energy levels of ³F₂, ³F₃, and ³H₄, since there is no radiational energy transfer between these levels. For the same reason, the n2 is to represent energy levels of ³F₄, and ³H₅. The excitation photon will be absorbed by sensitizer ions, pumping the electrons from n6 to n7. Then the energy at n7 is transferred to the emitter for both upconverting and down converting emission. There are three major energy transfer processes simultaneously happening inside a UCNP. The first process is the energy transferring from sensitizers (Yb³⁺) to emitters (Tm³⁺), and the transferring strength is characterized by transferring rate *C* (e.g. *C*₁, *C*₂, *C*₃, *C*₄). The second process is the carrier decaying on each of excited energy levels. Here we use *W* (e.g. *W*₂, *W*₃) to represent the decay rate and use *b* (e.g. *b*₅₄, *b*₅₃) to represent branching ratio. The third process is the cross-relaxation which plays a critical role for highly doped upconversion nanoparticles^{4,5}.

The rate equations to describe the system is shown as:

$$\frac{dn_1}{dt} = -c_1 n_1 n_7 + b_{21} w_2 n_2 + b_{31} w_3 n_3 + b_{41} w_4 n_4 + b_{51} w_5 n_5 - k_{41} n_1 n_4 - k_{31} n_1 n_3 - k_{51} n_1 n_3 \quad (1)$$

$$\frac{dn_2}{dt} = c_1 n_1 n_{S7} - c_2 n_2 n_7 - b_{21} w_2 n_2 + b_{32} w_3 n_3 + b_{42} w_4 n_4 + b_{52} w_5 n_5 + k_{41} n_1 n_4 + 2k_{31} n_1 n_3 \quad (2)$$

$$\frac{dn_3}{dt} = c_2 n_2 n_{S2} - c_3 n_3 n_7 - w_3 n_3 + b_{43} w_4 n_4 + b_{53} w_5 n_5 + 2k_{51} n_5 n_1 + k_{41} n_4 n_1 - k_{31} n_3 n_1 \quad (3)$$

$$\frac{dn_4}{dt} = c_3 n_3 n_7 - c_4 n_4 n_7 - w_4 n_4 + b_{54} w_5 n_5 - k_{41} n_1 n_4 \quad (4)$$

$$\frac{dn_5}{dt} = c_4 n_4 n_7 - w_5 n_5 - k_{51} n_1 n_5 \quad (5)$$

$$\frac{dn_7}{dt} = \frac{P\delta}{h\nu} n_6 - w_7 n_7 - (c_1 n_1 + c_2 n_2 + c_3 n_3 + c_4 n_4) n_7 \quad (6)$$

Here δ is the absorption cross-section, P is the excitation power, ν is the frequency of the excitation photon. The Plasmon enhancement mechanism for Yb³⁺ and Er³⁺ co-doped upconverting system has been developed by Dawei Lu et al.⁶. The mechanism for Yb³⁺ and Tm³⁺ co-doped upconverting system is similar. The metal surface that closes to a UCNP provides both plasmon enhancement and quenching effects. According to Dawei Lu et al. and Christian Clarke et al.⁷, the quenching effect will induce a large reduction in decay time. In our experiment, due to the larger spacing (>71.6 nm) the quenching effect is minimized, which is proved by the lifetime measurement shown in Figure S6. The plasmon enhancement will enhance the absorption cross-section (δ), the energy transferring rate (*C*) and the decay rate (*W*). In our experimental condition, we are using the excitation power density of 0.95 MW/cm², which achieves saturation condition⁸. Hence the enhancing in δ will not affect the energy transferring process. According to Dawei Lu et al. and Dung H. T. et al.⁹, when the distance is larger than $0.1\lambda_{sp}$ the enhancement on energy transferring can be ignored. Here the surface plasmon wavelength λ_{sp} is about 243.8 nm, and the distance between UCNP and film is larger than 71.6 nm that $> 0.1\lambda_{sp}$. Hence the enhancement on energy transferring rate can be ignored.

We now analyse the Purcell effect induced change on the emission rate and quantum yield. In the upconverting system, under strong excitation limit, without considering the cross-relaxation effect, the emission rate and the quantum yield will not be affected by the Purcell effect¹⁰, as the branching ratio is independent of the Purcell effect. Then the Purcell effect modulated equations are:

$$\begin{aligned} \frac{dn_1}{dt} &= -c_1 n_1 n_7 + P_{u21} w_2 n_2 + b_{31}(b_{31} P_{u31} + b_{32} P_{u32}) w_3 n_3 + b_{41}(b_{41} P_{u41} + b_{42} P_{u42} + b_{43} P_{u43}) w_4 n_4 + \\ & b_{51}(b_{51} P_{u51} + b_{52} P_{u52} + b_{53} P_{u53} + b_{54} P_{u54}) w_5 n_5 - k_{41} n_1 n_4 - k_{31} n_1 n_3 - k_{51} n_1 n_3 \\ \frac{dn_2}{dt} &= c_1 n_1 n_7 - c_2 n_2 n_7 - b_{21} P_{u21} w_2 n_2 + b_{32}(b_{31} P_{u31} + b_{32} P_{u32}) w_3 n_3 + b_{42}(b_{41} P_{u41} + b_{42} P_{u42} + \\ & b_{43} P_{u43}) w_4 n_4 + b_{52}(b_{51} P_{u51} + b_{52} P_{u52} + b_{53} P_{u53} + b_{54} P_{u54}) w_5 n_5 + k_{41} n_1 n_4 + 2k_{31} n_1 n_3 \\ \frac{dn_3}{dt} &= c_2 n_2 n_7 - c_3 n_3 n_7 - (b_{31} P_{u31} + b_{32} P_{u32}) w_3 n_3 + b_{43}(b_{41} P_{u41} + b_{42} P_{u42} + b_{43} P_{u43}) w_4 n_4 + \\ & b_{53}(b_{51} P_{u51} + b_{52} P_{u52} + b_{53} P_{u53} + b_{54} P_{u54}) w_5 n_5 + 2k_{51} n_5 n_1 + k_{41} n_4 n_1 - k_{31} n_3 n_1 \\ \frac{dn_4}{dt} &= c_3 n_3 n_7 - c_4 n_4 n_7 - (b_{41} P_{u41} + b_{42} P_{u42} + b_{43} P_{u43}) w_4 n_4 + b_{54}(b_{51} P_{u51} + b_{52} P_{u52} + b_{53} P_{u53} + \\ & b_{54} P_{u54}) w_5 n_5 - k_{41} n_1 n_4 \\ \frac{dn_5}{dt} &= c_4 n_4 n_7 - (b_{51} P_{u51} + b_{52} P_{u52} + b_{53} P_{u53} + b_{54} P_{u54}) w_5 n_5 - k_{51} n_1 n_5 \\ \frac{dn_7}{dt} &= \frac{P\delta}{h\nu} n_6 - P_{u76} w_7 n_7 - (c_1 n_1 + c_2 n_2 + c_3 n_3 + c_4 n_4) n_7 \end{aligned}$$

Here P_{ujk} (e.g. P_{u31} , $j = 3, k = 1$) is the Purcell factor for dipoles which have resonance frequency matches with the energy gap between level j and k . The P_{ujk} is either for the dipoles parallel to the mirror surface (horizontal dipoles) or perpendicular to the surface (vertical dipoles), for the case of calculating the enhancement for x/y dipoles and z dipoles respectively. We calculate the Purcell factors according to Chen et al.¹¹ As an example, Figure S7 shows the calculated P_{u52} (related to 455 nm emission, Figure S7a) and P_{u31} (related 800 nm emission, Figure S7b) for different spacing between the centre of the particle and the mirror surface.

From the plasmon modified equations, the change in decay rate from the Purcell effect will modify the steady-state carrier distribution, which will affect the emission intensity and the quantum yield. Based on the rate equation, the 455 nm emission photon number for unit time is $b_{52}(b_{51} P_{u51} + b_{52} P_{u52} + b_{53} P_{u53} + b_{54} P_{u54}) w_5 n_5$. Here n_5 is the carrier concentration at the energy level 1D_2 . The emission quantum yield for emission 455 nm is $b_{52}(b_{51} P_{u51} + b_{52} P_{u52} + b_{53} P_{u53} + b_{54} P_{u54}) w_5 / ((b_{51} P_{u51} + b_{52} P_{u52} + b_{53} P_{u53} + b_{54} P_{u54}) w_5 + k_{51} \times n_1)$. Figure S8 and S9 show the 455nm enhancement processes for vertical and horizontal orientated dipoles, respectively. For each of spacing value, there is one suit of steady-state distribution at each level. Figure S8a and S9a are the enhancement ratio for the emission, ratio=1 stands for the emission is the same as that with no enhancement. This factor depends on both Purcell factors and the steady-state concentration n_5 (Figure S8b and S9b). This emission enhancement ratio for different spacings is supposed to modify the weighting of the dipoles' amplitudes during our PSF simulation. Here this enhancement ratio is negligible. Figure S8c and S9c are the enhancement ratio for the emission (455 nm) quantum yield for vertical and horizontal orientated dipoles, respectively. This ratio depends on both Purcell factors and the steady-state concentration n_1 (Figure S8d and S9d).

Figure S5. Diagram of energy transfer processes of a Yb^{3+} and Tm^{3+} co-doped upconverting system.

Figure S6. (a) The measured 455nm fluorescence lifetime of UCNP for different spacings between the particle's centre and the mirror's surface. (b) The measured 455 nm fluorescence decay curves

Figure S7. Computed Purcell factor as a function of spacing (or distance of UCNP to mirror) with (a) 455 nm and (b) 800 nm emission wavelength for emitters oriented parallel or orthogonal to the mirrors' surface.

Figure S8. The enhancement for vertically orientated dipoles. (a) Enhancement ratio for the 455 nm emission. (b) The carrier distribution at the energy level of n_5 . (c) Enhancement ratio for the 455 nm emission quantum yield. (d) The carrier distribution at the energy level of n_1 .

Figure S9. The enhancement for horizontally orientated dipoles. (a) Enhancement ratio for the 455 nm emission. (b) The carrier distribution at the energy level of n_5 . (c) Enhancement ratio for the 455 nm emission quantum yield. (d) The carrier distribution at the energy level of n_1 .

Fourthly, the authors state on lines 120 to 122 that "the results from lifetime measurements (Figure S4a) and the analytic calculations of density of states (Figure S4b) confirms the plasmonic coupling is negligible with the working range larger than 75 nm". However, such statements are not supported by Figures S4a and S4b. On the contrary, Fig. S4b shows that the LDOS parallel to the mirror decreases by not less than 40% when the particle-to-mirror distance varies from 130 to 280 nm; same observation for the LDOS orthogonal to the mirror, comparing distances of 80 and 200 nm. In addition, the lifetime measurements shown in Fig. S4a are averaged over x , y , and z -orientations of the dipoles and, thus, they do not show anything on the lifetime variations for each of the dipole orientations. From theory, non-negligible variations are expected versus the distance to the metallic surface for the in-plane and out-of-plane dipole orientations; yet, since these variations do not follow the same trend and more or less compensate each other, they are not visible in the experimental lifetime measurements, where the contributions from all dipole orientations are averaged.

According to the comment for the last comment, the change in LDOS will slightly modify the emission amplitude. The plasmonic coupling is not negligible. We have revised our statement as "The lifetime measurement results (Figure S6) confirm that the plasmonic quenching is negligible when the working range is larger than 75 nm. The numerical simulation of energy transferring (see Supplementary Information Note 3) indicates a small emission enhancement due to the plasmonic coupling, which has been considered during the simulation process."

Fifthly, Figure 2 reveals significant discrepancies between the simulations and the experimental data, both in the xy and yz cuts, which the authors do not comment. In particular, the z -distribution of the intensity and the radii of the rings in the doughnut-shaped spots in the xy cuts do not match the model. The authors state on lines 160-161 that there is an "excellent agreement" between theory and experiment, which is not satisfying.

We appreciate the reviewer point this out. We agree that there are discrepancies between the simulations and the experimental result. We have revised our statement to "These experimental results qualitatively match the numerical simulation,". The discrepancies come from the small variations between experimental condition and simulation condition. Modelling of the PSF is a multi-parameter problem. The PSF depends on the optical wavelength, the emitter-to-mirror distance (modified Purcell factor and angular emission), immersion media, size of the nanoparticle (spatial coherence), the detected spectrum (temporal coherence), the quantum yield of the emitters, excitation condition (beam profile and excitation intensity), the optical setup itself (e.g. the numerical aperture and system aberration) and the photon-upconversion process. Here we compare the measurements with a simple model, three co-localized orthogonal dipoles radiating in front of a mirror with a single wavelength of emission, to capture the relevant physics. This model reproduces the main observations. We take this as an indication for the general applicability of this concept. In-depth discussion of the aforementioned effects is shown at the supplementary information.

We have added "Quantitatively modelling of the observed self-interferent emission PSF is a multi-parameter problem. The accuracy will depend on the parameters, such as the wavelength, the emitter-to-mirror distance (e.g. modified Purcell factor and angular emission), the immersion media, the size of the nanoparticle (spatial coherence), the detected spectrum (temporal coherence), the quantum yield of the emitters, the excitation condition (e.g. beam profile and intensity), the optics setup (e.g. the numerical aperture and system aberration) and the entire photon-upconversion process. Here we employ a simple model, three co-localized orthogonal dipoles radiating in front of a mirror with single wavelength of emission, to capture the

relevant physics, as this model can semi-quantitatively reproduce our experimental observations. We take this to illustrate the general applicability of the self-interference concept. For the in-depth discussions of the aforementioned effects, see the supplementary information Note 3.” into the main text to general explain the discrepancies.

More specifically, the discrepancies at the z-distribution of the intensity come from both the nonlinear photoresponse of the particle and the excitation condition. We generate the wide-field excitation beam by focusing a Gaussian beam at the back aperture of an objective lens. The method is a common method for total internal reflection fluorescence microscope (Biophysical Reviews, 04 May 2019, 11(3):319-325) and other wide-field excitations (Nature Photonics volume 12, pages548–553, 2018). However, the excitation field along the z-axis is not a constant, it has a small variation, as the method is similar to the case of defocusing a Gaussian beam. During our experiment, when we move the stage with a direction away from the mirror surface, the excitation intensity will slightly increase as the excitation beam is narrower at the sample plane. This effect will cause a longer depth of field at the z-distribution. Besides, UCNPs have a stable emission saturation effect, this effect will further increase the depth of field of z-distribution. Hence the experimental z-distribution is a little bit longer than the simulation result. We added “**The experimental result shows a longer depth of field at the z-axis distribution of the intensity, which is due to the non-parallel excitation of the wide-field beam.**” at the main text to clarify this discrepancy.

We apologize that the image scale for Figure 2a-2f is misleading; the scale bar is smaller than that in Figure 2A-2F. This, in turn, makes the doughnut spots look much smaller than the simulated result. We have revised Figure 2 by scaling all result with the same scale bar. Apart from the scale bar issue, there is small discrepancy at the shapes of 2D PSF, e.g. the radius of the rings, which comes from the system aberration. We added “**The slight discrepancy of 2D PSF shapes between the experimental and the simulated results are caused by the system aberration.**” at the main text to clarify this discrepancy. Besides, we found that the z-axis scale in Figure 2g-l, Figure S10k-t and Figure S11k-t are not correct. We have corrected them at the revised Figure 2, Figure S10 and Figure S11.

(3) The methodology used to demonstrate the main claim of the manuscript (i.e., axial localization of emitters with 0.9 nm accuracy) is questionable. Indeed, it is based on two reasonings that may be easily contradicted. Firstly, Figure S10 shows that the authors have estimated the axial resolution of their axial localization technique from an extrapolation of the error bars on a few experimental points, by simply linking the points with a straight segment of line, which is not accurate. Indeed, fitting the experimental data with a model would be more accurate. In addition, the resulting estimation is much dependent on the definition of the measurement errors or fit uncertainties, whatever the way such error bars are obtained, which is arbitrary.

We agree that simply linking data points is not an optimal method. To address the reviewer’s concern, we used theoretical simulated results to fit the lines between data points, as shown in the revised Figure 3 and the revised Figure S17. According to the new calibration curve, we have revised the distance sensing result at the revised Figure 4. We have revised our statement on resolution as “ We demonstrate a real-time distance sensing technology with a localization **accuracy of 2.8 nm, according to the AFM characterization values, smaller than 1/350 of the excitation wavelength.**” in the abstract, “**According to the cross-validation method (Figure S21), the measured values for Step 1, 2 and 3 by our fluorescent self-interference method are 103.7 nm (Figure 4d), 124.7 nm (Figure 4d) and 130.8 nm (Figure 4e), respectively, suggesting that the differences between our method and the AFM values are smaller than 2.8 nm.**” and “ **A cross-validation method is used to evaluate distance by taking all the four characterization curves into the considerations, as shown in Supplementary Information note 5 and Figure S21. The estimated localization resolutions across the different distance ranges are shown in Figure S22, where the sub-5 nm resolution can be achieved for the**

spacing ranges of 104-137 nm, 160-186 nm and 282-365 nm (Figure S22c).” at the main text. We have revised the distance sensing result in Figure 4h-j, the changes are negligible.

For our method, the resolution is limited by the variation in the measured PSF, which is induced by both the slightly offset of nanoparticles on the flat surface and the non-uniformity of nanoparticles. This variation is much larger than the fitting induced variation. Hence, we use experimental variations as error bars for data points. The error for the points in between two data points is obtained by linearly linking errors from two data points. With the new calibration curve, as the points in between two data points are generated by simulation, the new errors can be found by interpolating error values according to the change of averaged value. The detail is shown at the revised Note 6 at the Supplementary information. According to the change of the calibration curve, we revised the analysis for resolution estimation at the revised Figure S22.

Figure 3 | Features and quantitative analysis of the series of observed lateral PSFs used for determining the distance between UCNP and mirror. (a-f) Cross-section line profiles of the lateral PSF of UCNP (shown as insets) for six different spacings. (g-j) The four characteristic parameters, including FWHM, iFWHM, Area, and Depth, respectively, measured as a function of the distance between UCNP and mirror. Additional PSF analysis results for 455 nm and 800 nm emission are provided in the supplementary information Figure S17 and Figure S18, respectively. Further details for resolution analysis are explained in the supplementary information note 5.

Figure S17. Variation of feature from the observed 455 nm emission PSF used to determine the distance between UCNPs and mirror. (a-j) Cross-section through the PSF of UCNPs (shown as insets) at spacing distances of 71.6 nm, 103.8 nm, 133.6 nm, 154.9 nm, 173.5 nm, 186.6 nm, 214.7 nm, 267.5 nm, 326.9 nm, and 483.9 nm. (k-n) The four characteristic parameters of the PSF, FWHM, iFWHM, Area, and Depth, respectively, measured as a function of the distance between UCNPs and mirror.

Figure S22. (a) Values for the localization precision of the derived spacing for four calibration curves (FWHM, Area, iFWHM, Depth). (b) The selection rule for obtaining best resolution values at the different spacing region. “1”, “2”, “3”

and “4” indicate the selection of FWHM, Area, iFWHM, and Depth, respectively. (c) Resulting best values for the localization precision by using the self-interference effect.

Secondly, the authors infer the axial resolution of their axial localization technique from a comparison to the thickness measurements of the spacing layer between the nanoparticles and the mirror obtained using an atomic force microscope. However, Figure 4 clearly suggests that the authors have averaged the measurements over a number of nanoparticles, which may strongly compensate for the error on each measurement and the spatial dispersion of the layer thickness due to the layer deposition technique and the inherent roughness of the metallic film, which may be higher than the claimed resolution of the authors' technique.

The result obtained by AFM is an averaged value from multiple positions. Hence the results by our method are comparable with the results by AFM. The thickness for each step is calculated by averaging the positions within a large area, as the averaged values are more reasonable to statistically represent the thickness for both AFM and our method. We have added this information into the revised caption of Figure 4, as: “**The heights are measured by averaging the multiple positions within region.**”

Here the sensing resolution (shown in Figure S22) is calculated according to the calibration curve (Figure 3g-j), which is limited by both the shape of the curve and the experimental measuring error. The measuring accuracy stands for the difference between the AFM measured averaged value and the averaged value by our method. This accuracy is not the resolution. To avoid confusion, we have revised the text in the abstract as “**We demonstrate a real-time distance sensing technology with a localization accuracy of 2.8 nm, according to the AFM characterization values, smaller than 1/350 of the excitation wavelength.**”

The latter demonstration shows that the authors' technique may be useful for estimations of the thickness of a layer with sub-nm accuracy, based on measurements on several nanoparticles deposited on the layer; however, this does not conclusively demonstrate that sub-nm resolution is attained for one individual nanoparticle or emitter.

The resolution estimation for the individual nanoparticle is shown at the revised Figure S22, where, the sub-5nm resolution can be achieved for the spacing range of 104-137 nm, 160-186 nm and 282-365 nm. The averaged result from many particles provides an averaged thickness value which could be close to the averaged thickness measured by AFM with difference smaller than resolution. To further address the reviewer's concern on the resolving power from a single nanoparticle, we have added one experiment to detect individual particles' height on a flat surface. As shown in Figure S24a, we put both 32.8 nm \pm 1.28 (diameter) and 41.67 nm \pm 1.54 (diameter) nanoparticles on a mirror substrate with 145 nm SiO₂ spacer. Here each of particles has the ability to sense the distance with a resolution of sub-5 nm. The measured results show the two groups of samples. The larger particles (40 nm) show measured distances from 166.2 nm to 169.4 nm labelled by warm colour, with an averaged value of 167.2 nm. The smaller particles (33 nm) show measured distances from 161.8 nm to 164.3 nm labelled by cold colours, with an averaged value of 163.2 nm. The difference between the averaged value of two particles matches with the SEM characterized size difference within the error range. Here the slighter larger size for 40 nm particle leads to a tiny bit change (0.36%-9%) in its calibration value, which has been considered. Figure S24b shows typical cross-sections of PSFs from two types of nanoparticles.

We added “ **We further verify the resolving power of the method by measuring the heights of two batches of nanoparticles on the same substrate. As shown in Figure S24, the height of each particle could be resolved, and the measured average diameter difference between two batches are 8 \pm 4.8 nm which is close to the**

difference of 8.9 ± 2.8 nm verified by transmission electron microscope (TEM) measurement (Figure S25).” into the main text.

Figure S24. Large field distance sensing of UCNPs. (a) The 2D fluorescence image of UCNPs on a mirror surface. The characterized thickness of the spacer is 145 nm. The UCNPs are from two batches of samples with averaged diameters of 32.8 ± 1.28 nm and 41.67 ± 1.54 nm. The averaged centre height for particle batch 1 and 2 are 163.2 ± 1 nm and 167.2 ± 1.4 nm respectively. (b) The cross-section of two typical UCNPs in (a). The measured distances from the particles’ centre to mirror surface are 166 nm and 163 nm. The SEM images and the size distribution of particles are shown in Figure S25.

Figure S25. TEM images (Top) and size distribution histograms (Bottom) of the nanoparticles. a) NaYF₄: 20% Yb, 4% Tm. (b) NaYF₄: 20% Yb, 4% Tm. (c) The size distribution of nanoparticles in (a). (d) The size distribution of nanoparticles in (b). Scale bar is 50 nm.

REVIEWER COMMENTS

Reviewer #1 (Remarks to the Author):

In the revision, the authors minutely response most of the questions or recommendations raised by the reviewers and made corresponding amendments or corrections point by point, which have significantly improved the quality of the manuscript. In particular, the discussions about surface plasmon effect, the requirement of NA and the range of sensing distance have been analyzed/added to make the study more credible. Besides, some theoretical analysis and experiments are added as well, such as detecting individual particles' height on a flat surface. And thus, I recommend the article in the current form is worth publishing in Nature Communications.

Reviewer #2 (Remarks to the Author):

Authors significantly improve the manuscript quality along with further experiments and theoretical explanations. They addressed the questions point by point. I recommend for publication after revising the following minor concern on the reflected phase explanations.

In the self-interference on a single surface, the field reflection ($r = |r| \exp(\varphi)$) is important other than reflectance ($|r|^2$). The phase change directly affects the self-interference pattern. The reflected phase φ depends on the refractive indices of the interfaces. Based on that, (Line 206), the authors claim that the reflected phase is π . This is only true for dielectric interfaces with only real refractive indices ($n_2 > n_1$ and n_2, n_1 are real numbers). The imaginary part of Ag's refractive index induces phases different than π . For example, From the Fresnel equations, the phase will be -139.35° for $n_1 = 1$ and $n_2 = 0.0420 + 2.6994i$ with zero incident angle. Different mediums will affect this phase as well. The phases introduced in water ($n_1 = 1.33$) and oil ($n_1 = 1.52$) are -127.54° and -121.24° .

Please revise your phase discussion in terms of field reflection instead of intensity reflection.

Reviewer #3 (Remarks to the Author):

In the revised version of their manuscript, Liu et al have added a fairer overview of the state of the art, solved previous inconsistencies in the theoretical part, and added a more accurate estimation of their technique's performance. Thus, I recommend that this revised version can be accepted for publication in Nature Communications.

Point by point response to the reviewers' comments:

Reviewer #2 (Comments for the Author):

Authors significantly improve the manuscript quality along with further experiments and theoretical explanations. They addressed the questions point by point. I recommend for publication after revising the following minor concern on the reflected phase explanations.

In the self-interference on a single surface, the field reflection ($r = |r| \exp(\varphi)$) is important other than reflectance ($|r|^2$). The phase change directly affects the self-interference pattern. The reflected phase φ depends on the refractive indices of the interfaces. Based on that, (Line 206), the authors claim that the reflected phase is π . This is only true for dielectric interfaces with only real refractive indices ($n_2 > n_1$ and n_2 and n_1 are real numbers). The imaginary part of Ag's refractive induces phases different than π . For example, From the Fresnel equations, the phase will be -139.35 for $n_1 = 1$ and $n_2 = 0.0420 + 2.6994i$ with zero incident angle. Different mediums will affect this phase as well. The phases introduced in water ($n_1 = 1.33$) and oil ($n_1 = 1.52$) are -127.54 and -121.24 .

Please revise your phase discussion in terms of field reflection instead of intensity reflection.

We sincerely thank the reviewer for pointing this out. We have revised the phase discussion part as shown “The field reflection of the emission at the interface between the beam propagating medium (SiO_2) and the silver (Ag) surface induces an external phase shift φ_{refl} , with an expression of $\varphi_{refl} = \pi - \tan^{-1}((n - n_{ag})/(n + n_{ag}))$ for a normal incident angle, according to the Fresnel law. Here n and n_{ag} are the complex refractive indices of the propagating medium and the silver mirror, respectively. ”. We have fixed a mistake in equation 2, where the Gouy phase shift φ_{Gouy} should be $\tan^{-1} \frac{2\Delta z}{z_R}$, rather than $-\tan^{-1} \frac{2\Delta z}{z_R}$. We have recalculated the phase difference values that are for 1.9π and 3.1π for the spacing values of 71.6 nm and 154.9 nm, respectively. We have updated these values in the main text.

Here we use equation 1 and 2 to explain and analyze the self-interference effect. We use a finite-element method (Note 2) to deliver the numerical simulation; this method embeds a correct phase term. Hence, all simulation results were accurate. The different medium should affect the reflection phase shift. We have included this change in the phase shift in Figure S12 and S13, as the simulation method embeds this phase change.

REVIEWERS' COMMENTS

Reviewer #2 (Remarks to the Author):

The authors addressed the minor concern on the reflected field phase. With that, I recommend for publication at Nature Communications.